# No-Regret Learning in Harmonic Games: Extrapolation in the Face of Conflicting Interests

**Davide Legacci**[*]     **Panayotis Mertikopoulos**
Univ. Grenoble Alpes, CNRS, Inria, Grenoble INP, LIG 38000 Grenoble, France
{davide.legacci,panayotis.mertikopoulos}@univ-grenoble-alpes.fr

**Christos Papadimitriou**          **Georgios Piliouras**
Columbia University, NYC, &          Google DeepMind
Archimedes/Athena RC, Greece          London, UK
christos@columbia.edu          gpil@google.com

**Bary Pradelski**
CNRS, Maison Française d'Oxford
2–10 Norham Road, Oxford, OX2 6SE, United Kingdom
bary.pradelski@cnrs.fr

## Abstract

The long-run behavior of multi-agent learning – and, in particular, *no-regret learning* – is relatively well-understood in potential games, where players have aligned interests. By contrast, in harmonic games – the strategic counterpart of potential games, where players have *conflicting* interests – very little is known outside the narrow subclass of 2-player zero-sum games with a fully-mixed equilibrium. Our paper seeks to partially fill this gap by focusing on the full class of (generalized) harmonic games and examining the convergence properties of follow-the-regularized-leader (FTRL), the most widely studied class of no-regret learning schemes. As a first result, we show that the continuous-time dynamics of FTRL are *Poincaré recurrent*, that is, they return arbitrarily close to their starting point infinitely often, and hence fail to converge. In discrete time, the standard, "vanilla" implementation of FTRL may lead to even worse outcomes, eventually trapping the players in a perpetual cycle of best-responses. However, if FTRL is augmented with a suitable extrapolation step – which includes as special cases the optimistic and mirror-prox variants of FTRL – we show that learning converges to a Nash equilibrium from any initial condition, and all players are guaranteed at most $\mathcal{O}(1)$ regret. These results provide an in-depth understanding of no-regret learning in harmonic games, nesting prior work on 2-player zero-sum games, and showing at a high level that harmonic games are the canonical complement of potential games, not only from a strategic, but also from a dynamic viewpoint.

## 1   Introduction

The question of "as if" rationality – that is, whether selfishly-minded, myopic agents may learn to behave "*as if*" they were fully rational – has been one of the cornerstones of non-cooperative game theory, and for good reason. Especially in modern applications of game theory to machine learning and data science – from online ad auctions to recommender systems and multi-agent reinforcement

---

[*]Corresponding author.

38th Conference on Neural Information Processing Systems (NeurIPS 2024).

learning – the standard postulates of rationality (knowledge of the game, capacity to compute an equilibrium, flawless execution of equilibrium strategies, common knowledge of rationality, etc.) are almost never met in practice; as a result, game-theoretic predictions that rely on these assumptions are likewise put into question. By contrast, given the ease of implementing and deploying cheap, computationally efficient learning algorithms and policies at a large scale, it is often more logical to turn to the policy being deployed as the object of interest. The aim is then to understand its long-run behavior – and, in particular, whether it ultimately leads to equilibrium.

A major obstacle in this approach is the complexity of computing a Nash equilibrium, a problem which is known to be complete for PPAD – and hence intractable – by the seminal work of Daskalakis et al. [12]. This result implies that it is not plausible to expect any algorithm to converge to Nash equilibrium in *all* games (at least, not in a reasonable amount of time), so it dovetails naturally with the impossibility results of Hart & Mas-Colell [22, 23] who showed that there are no uncoupled learning dynamics that converge to Nash equilibrium in all games. On that account, it is natural to ask in which classes of games we can expect a learning algorithm to converge, in which classes we cannot, and under what conditions.

Perhaps the most well-behaved class of games in terms of learning is the class of *potential games* [44, 55], where players have *common* interests – not necessarily driving them to play the same strategy, but in the sense that externalities are symmetric and aligned along a common objective (the potential of the game). In this class of games, the behavior of learning dynamics – and, in particular, no-regret learning [8, 13, 19, 27, 28, 36, 40, 43, 59] – are relatively well understood, and there is a wide range of equilibrium convergence results, from continuous to discrete time, and even with bandit, payoff-based feedback [24, 25, 55].

By contrast, in the presence of *conflicting* interests, the situation can be quite different. In two-player zero-sum games with a fully-mixed equilibrium – such as Matching Pennies – the continuous-time dynamics of no-regret, regularized learning are recurrent in the sense of Poincaré – that is, the induced trajectory of play returns arbitrarily close to where it started infinitely many times [41, 48]. In discrete time, the situation becomes more complicated: the vanilla version of follow-the-regularized-leader (FTRL) – the most widely studied family of no-regret algorithms – is no longer recurrent, but it diverges away from equilibrium in the same class of games [18, 42]. On the other hand, if players employ an optimistic / extra-gradient variant of FTRL, the induced trajectory of play converges to equilibrium [15, 42] and, under certain conditions, it is even possible to show that it converges at a geometric rate [62].

At the same time, zero-sum games may also admit a potential function, so it is not possible to predict the outcome of a learning process based on where it stands along the potential / zero-sum axis. The non-trivial intersection of these classes means that potential and zero-sum games are *not* complementary, and this, not only from a strategic, but also from a dynamic viewpoint. Instead, the true strategic complement of potential games is the class of *harmonic games*. This class was first considered by Candogan et al. [6], who established a remarkable decomposition result: Every game in normal form can be decomposed as the sum of a potential game and a harmonic game, and this decomposition is unique up to affine transformations that do not alter the equilibrium outcomes of the game. In particular, the class of potential and harmonic games intersect trivially (up to strategic equivalence), and all two-player zero-sum games with an interior equilibrium are harmonic, thus lending credence to the fact that it is harmonic games, not zero-sum games, that correctly capture the notion of conflicting interests in this context.

This raises the question:

*What is the behavior of no-regret algorithms and dynamics in harmonic games?*

Except for a very recent paper by Legacci et al. [35] (which we discuss below), almost nothing is known on this question. With this backdrop, our contributions can be summarized as follows:

1. Starting with a continuous-time model of no-regret learning, we show that all FTRL dynamics are Poincaré recurrent in all harmonic games. This generalizes and extends the recent result of Legacci et al. [35] for the replicator dynamics in uniform harmonic games (a subclass of harmonic games in which the uniform distribution is always a Nash equilibrium).[2]

---

[2]In more detail, the way that Legacci et al. [35] obtained their result hinges on the so-called *Shahshahani metric*, a choice which is essentially "mandated" by the structure of the replicator dynamics. Specifically, the key

2. In discrete-time models of learning, the standard implementation of FTRL cannot be expected to converge (since it fails to do so in Matching Pennies). To correct this behavior, we consider a flexible algorithmic template, inspired by Azizian et al. [3] and dubbed *extrapolated FTRL* (FTRL+), which augments FTRL with a forward-looking, extrapolation step (including as special cases the optimistic and extra-step variants of FTRL, cf. Section 4). We then establish the following results:

   (a) Under extrapolated FTRL, players are guaranteed constant individual regret (so, as a consequence, the players' empirical frequency of play converges to coarse correlated equilibrium at a rate of $\mathcal{O}(1/T)$).[3] This should be contrasted with the results of [13, 14] who showed that players can achieve *polylogarithmic* regret in any game (finite or convex).

   (b) The induced trajectory of play converges to Nash equilibrium from any initial condition.

Our results aim to provide an in-depth understanding of no-regret learning in harmonic games, nesting prior work on 2-player zero-sum games – from Poincaré recurrence [41, 48] to constant regret [27] and convergence under optimistic / extra-gradient schemes [11, 15, 18, 42, 62]. In partiucular, at a high level, our results show that harmonic games are the canonical complement of potential games, not only from a strategic, but also from a dynamic, learning viewpoint.

## 2  Preliminaries

**2.1.  Preliminaries on finite games.** Throughout the sequel, we will work with *finite games in normal form*. Formally, such games consist of (*i*) a finite set of *players* $i \in \mathcal{N} \equiv \{1, \ldots, N\}$; (*ii*) a finite set of *actions* $\mathcal{A}_i$ per player $i \in \mathcal{N}$; and (*iii*) an ensemble of *payoff functions* $u_i \colon \prod_j \mathcal{A}_j \to \mathbb{R}$, each determining the reward $u_i(\alpha)$ of player $i \in \mathcal{N}$ in a given action profile $\alpha = (\alpha_1, \ldots, \alpha_N)$. Putting everything together, we will write $\mathcal{A} \coloneqq \prod_i \mathcal{A}_i$ for the game's *action space* and $\Gamma \equiv \Gamma(\mathcal{N}, \mathcal{A}, u)$ for the game with primitives as above.

During play, each player selects an action according to some *mixed strategy*, that is, a probability distribution $x_i$ over $\mathcal{A}_i$ which assigns probability $x_{i\alpha_i}$ to $\alpha_i \in \mathcal{A}_i$. In a slight abuse of notation, if $x_i$ assigns all probability mass to some action $\alpha_i \in \mathcal{A}_i$ (that is, $x_{i\alpha_i} = 1$), we will identify $x_i$ with $\alpha_i$ and we will call it *pure*. We will also write $\mathcal{X}_i \coloneqq \Delta(\mathcal{A}_i) \subseteq \mathbb{R}^{\mathcal{A}_i}$ for the mixed strategy space of player $i$, $x = (x_1, \ldots, x_N)$ for the *strategy profile* collecting the strategies of all players, and $\mathcal{X} \coloneqq \prod_i \mathcal{X}_i$ for the game's *strategy space*.

The *mixed payoff* of player $i$ under a mixed strategy profile $x \in \mathcal{X}$ may then be written as

$$u_i(x) = \mathbb{E}_{\alpha \sim x}[u_i(\alpha)] = \sum_{\alpha \in \mathcal{A}} u_i(\alpha)\, x_\alpha = \sum_{\alpha_i \in \mathcal{A}_i} u_i(\alpha_i; x_{-i})\, x_{i\alpha_i} \tag{1}$$

where $x_\alpha \coloneqq \prod_i x_{i\alpha_i}$ denotes the joint probability of $\alpha = (\alpha_1, \ldots, \alpha_N) \in \mathcal{A}$ under $x \in \mathcal{X}$, and, in standard game-theoretic notation, we write $(x_i; x_{-i}) = (x_1, \ldots, x_i, \ldots, x_N)$ for the profile where player $i$ plays $x_i \in \mathcal{X}_i$ against the strategy $x_{-i} \in \mathcal{X}_{-i} \coloneqq \prod_{j \neq i} \mathcal{X}_j$ of all other players. We also respectively define the *individual payoff field* of player $i$ and the *game's payoff field* as

$$v_i(x) = (u_i(\alpha_i; x_{-i}))_{\alpha_i \in \mathcal{A}_i} \quad \text{and} \quad v(x) = (v_1(x), \ldots, v_N(x)) \tag{2}$$

so $u_i(x) = \sum_{\alpha_i \in \mathcal{A}_i} v_{i\alpha_i}(x)x_{i\alpha_i} \equiv \langle v_i(x), x_i \rangle$, where $\langle \cdot, \cdot \rangle$ is the standard duality pairing on $\mathbb{R}^{\mathcal{A}_i}$. By multilinearity, each player's individual payoff field is Lipschitz continuous on $\mathcal{X}$, and we will write $G_i$ for its Lipschitz modulus, that is

$$\|v_i(x') - v_i(x)\|_* \leq G_i \|x' - x\| \quad \text{for all } x, x' \in \mathcal{X}. \tag{3}$$

*Remark.*  In the above and throughout, $\|\cdot\|$ denotes an ambient norm on $\mathbb{R}^{\mathcal{A}_i}$ (usually the $L^1$ norm), and $\|\cdot\|_*$ is the corresponding dual norm (usually the $L^\infty$ norm). To simplify notation, we will not carry the player index $i$ in $\|\cdot\|$, and we will instead rely on the context to resolve any ambiguities.

---

property of the Shahshahani metric is that incompressibility of the replicator field is equivalent to the underlying game being uniformly harmonic; however, finding a variant of the Shahshahani metric attuned to FTRL seems to be a formidable task, and likewise for non-uniform harmonic games. Because of this, the "incompressibility" approach of [35] does not seem applicable to our setting – at least, not in a straightforward way.

[3] We clarify here that "constant" refers to the horizon $T$ of play; the dependence on the number of actions may be logarithmic or worse (depending on the specific regularized learning scheme employed by the players).

In terms of solution concepts, we will focus almost exclusively on the notion of a *Nash equilibrium* (NE), i.e., a strategy profile $x^* \in \mathcal{X}$ that is unilaterally stable in the sense that

$$u_i(x^*) \geq u_i(x_i; x^*_{-i}) \quad \text{for all } x_i \in \mathcal{X}_i, i \in \mathcal{N}. \tag{NE}$$

Equivalently, (NE) can be expressed in terms of the game's payoff field as a variational inequality of the form

$$\langle v(x^*), x - x^* \rangle \leq 0 \quad \text{for all } x \in \mathcal{X}. \tag{VI}$$

Thus, writing $\mathrm{supp}(x^*_i) = \{\alpha_i \in \mathcal{A}_i : x_{i\alpha_i} > 0\}$ for the *support* of $x^*_i$, it follows that $x^*$ is a Nash equilibrium if and only if $u_i(\alpha_i; x^*_{-i}) \geq u_i(\beta_i; x^*_{-i})$ for all $\alpha_i \in \mathrm{supp}(x^*_i)$ and all $\beta_i \in \mathcal{A}_i, i \in \mathcal{N}$. We will use all this freely in the rest of our paper.

## 2.2. Harmonic games.
Our main focus in what follows will be the class of *harmonic games*, first introduced by Candogan et al. [6] as a game-theoretic model for strategic situations with conflicting, anti-aligned interests. Specifically, as was shown by Candogan et al. [6] – and, in a more general setting, by Abdou et al. [1] – every game in normal form can be decomposed as the sum of a potential game and a harmonic game, and this decomposition is unique up to affine transformations that do not alter the equilibrium outcomes of the game.[4] In this decomposition, the potential component of a game captures multi-agent strategic interactions with *common* interests, whereas the harmonic component covers interactions with *conflicting* interests.[5]

Formally, adapting the more general setup by Abdou et al. [1], we have the following definition:

**Definition 1.** A finite game $\Gamma \equiv \Gamma(\mathcal{N}, \mathcal{A}, u)$ is said to be *harmonic* when it admits a *harmonic measure*, i.e., a collection of weights $\mu_{i\alpha_i} \in (0, \infty)$, $\alpha_i \in \mathcal{A}_i, i \in \mathcal{N}$, such that

$$\sum_{i \in \mathcal{N}} \sum_{\beta_i \in \mathcal{A}_i} \mu_{i\beta_i} [u_i(\alpha_i; \alpha_{-i}) - u_i(\beta_i; \alpha_{-i})] = 0 \quad \text{for all } \alpha \in \mathcal{A}. \tag{HG}$$

In particular, if $\Gamma$ is harmonic relative to the uniform measure $\mu_{i\alpha_i} = 1$, $\alpha_i \in \mathcal{A}_i, i \in \mathcal{N}$, we will say that $\Gamma$ is a *uniform harmonic game* (UHG).

*Remark.* With regard to terminology, Candogan et al. [6] call "harmonic games" what we call "uniform harmonic games", and Abdou et al. [1] call "$\mu$-harmonic games" what we call "harmonic games".[6] We use this convention because it simultaneously simplifies notation and terminology while capturing all relevant strategic features of the game; for a detailed discussion, see Appendix A. To avoid needless repetition, and unless there is a danger of confusion, when we say that $\Gamma$ is harmonic, we will write $\mu_i$ for the corresponding measure, and we will write $m_i = |\mu_i| = \sum_{\beta_i \in \mathcal{A}_i} \mu_{i\beta_i}$ for the total mass of $\mu_i$. ◇

Broadly speaking, in harmonic games, for any player considering a deviation toward a specific pure strategy profile, there exist other players with an incentive to deviate *away* from said profile. In this regard, harmonic games can be seen as the strategic complement of potential games, where player interests are aligned and sequences of unilateral best responses generate a finite improvement path that terminates at a pure Nash equilibrium [44]. By contrast, except for trivial cases (like the zero game) harmonic games *do not* admit pure Nash equilibria, and they possess non-terminating best-response paths. For all these reasons, harmonic games can be considered as "orthogonal" to potential games, in a sense made precise by the decomposition results of Candogan et al. [6] and Abdou et al. [1].

It is of course natural to ask what is the relation between harmonic games and zero-sum games. Games belonging to the latter class – such as Matching Pennies and Rock-Paper-Scissors – have long been used as prototypical examples of strategic conflict; at the same time, there are zero-sum games that are also potential (and even possess strict equilibria), so the potential / zero-sum distinction does not capture the whole picture. As a matter of fact, it is not a coincidence that the textbook examples of zero-sum games admit fully-mixed Nash equilibria: as we discuss in Appendix A, two-player zero-sum games with a fully mixed Nash equilibrium are harmonic, so the existing results for such games are, in a sense, more closely attuned to their harmonic character.

---

[4]We briefly recall here that $\Gamma \equiv \Gamma(\mathcal{N}, \mathcal{A}, u)$ is a potential game if it admits a *potential function* $\phi \colon \mathcal{X} \to \mathbb{R}$ such that $u_i(\beta_i; \alpha_{-i}) - u_i(\alpha_i; \alpha_{-i}) = \phi(\beta_i; \alpha_{-i}) - \phi(\alpha_i; \alpha_{-i})$ for all $\alpha, \beta \in \mathcal{A}$ and all $i \in \mathcal{N}$ [44].

[5]The terminology "harmonic" is due to Candogan et al. [6] and alludes to the harmonic component of the graphical Hodge decomposition [30].

[6]To be even more precise, the definition of Abdou et al. [1] involves an additional set of weights, called a *comeasure*; however, as we explain in Appendix A, these weights do not change the preference structure of the game, so we disregard this extra degree of generality.

# 3 Continuous-time analysis: Poincaré recurrence

The most basic rationality postulate in the context of online learning is the minimization of a player's (external) regret, i.e., the difference between a player's cumulative payoff and that of the player's best possible strategy in hindsight. In more detail, assuming for the moment that play evolves in continuous time, the *regret* of player $i \in \mathcal{N}$ relative to a sequence of play $x(t) \in \mathcal{X}$ is defined as

$$\text{Reg}_i(T) = \max_{p_i \in \mathcal{X}_i} \int_0^T [u_i(p_i; x_{-i}(t)) - u_i(x(t))] \, dt \tag{4}$$

and we say that the player has *no regret* under $x(t)$ if $\text{Reg}_i(T) = o(T)$ as $T \to \infty$.

The most widely used scheme for attaining no regret is the family of policies known as *follow-the-regularized-leader* (FTRL) [57, 58]. At a high level, the idea behind FTRL is that, at all times $t \geq 0$, each player $i \in \mathcal{N}$ plays a mixed strategy $x_i(t) \in \mathcal{X}_i$ that maximizes the player's cumulative payoff up to time $t$ minus a certain regularization penalty. In our continuous-time setting, this gives rise to the FTRL dynamics

$$x_i(t) = \arg\max_{p_i \in \mathcal{X}_i} \left\{ \int_0^t u_i(p_i; x_{-i}(\tau)) \, d\tau - h_i(p_i) \right\} = \arg\max_{p_i \in \mathcal{X}_i} \left\{ \int_0^t \langle v_i(x(\tau)), p_i \rangle \, d\tau - h_i(p_i) \right\} \tag{5}$$

or, more compactly,

$$\dot{y}_i(t) = v_i(x(t)) \qquad x_i(t) = Q_i(y_i(t)) \tag{FTRL-D}$$

where $h_i \colon \mathcal{X}_i \to \mathbb{R}$ is a convex penalty function known as the *regularizer* of the method, $Q_i$ denotes the *regularized choice map* of player $i$, and $Q = (Q_1, \ldots, Q_N)$ denotes the profile thereof. Formally, writing $\mathcal{Y}_i \equiv \mathbb{R}^{A_i}$ for the *payoff space* of player $i \in \mathcal{N}$ – that is, the space of all possible payoff vectors $v_i$ of player $i$ – the regularized choice map $Q_i \colon \mathcal{Y}_i \to \mathcal{X}_i$ is defined as

$$Q_i(y_i) = \arg\max_{x_i \in \mathcal{X}_i} \{\langle y_i, x_i \rangle - h_i(x_i)\} \quad \text{for all } y_i \in \mathcal{Y}_i. \tag{6}$$

In essence, $Q_i$ is a "soft" version of the arg max correspondence $y_i \mapsto \arg\max_{x_i \in \mathcal{X}_i} \langle y_i, x_i \rangle$, suitably regularized by a penalty term intended to incentivize exploration. For technical reasons, we will also assume that each $h_i$ is *strongly convex*, i.e.,

$$h_i(tx_i + (1-t)x_i') \leq th_i(x_i) + (1-t)h_i(x_i') - \tfrac{1}{2}K_i t(1-t)\|x_i - x_i'\|^2 \tag{7}$$

for some $K_i > 0$ (commonly referred to as the *strong convexity modulus* of $h_i$), and for all $x_i, x_i' \in \mathcal{X}$, $t \in [0, 1]$. In plain words, this simply means that $h_i$ has "enough curvature" in the sense that it can be bounded from below by a (positive) quadratic function which agrees with $h_i$ to first order.

The go-to example of this setup is the entropic regularizer

$$h_i(x_i) = \sum_{\alpha_i \in \mathcal{A}_i} x_{i\alpha_i} \log x_{i\alpha_i} \tag{8}$$

which yields the so-called *logit choice map*

$$Q_i(y_i) \equiv \Lambda_i(y_i) := \frac{(\exp(y_{i\alpha_i}))_{\alpha_i \in \mathcal{A}_i}}{\sum_{\alpha_i \in \mathcal{A}_i} \exp(y_{i\alpha_i})} \quad \text{for all } y_i \in \mathcal{Y}_i. \tag{9}$$

By Pinsker's inequality, the entropic regularizer is 1-strongly convex relative to the $L^1$-norm on $\mathcal{X}_i$ [57], and by a standard calculation [37, 54], the induced sytem (FTRL-D) boils down to the replicator dynamics of Taylor & Jonker [60]. Some other standard examples of (FTRL-D) include the Euclidean projection dynamics of Friedman [17] when $h_i(x_i) = (1/2)\|x_i\|_2^2$, the $q$-replicator dynamics [21, 38], etc. To streamline our presentation, we defer a detailed discussion of these examples to Appendix C, and we proceed below to state the main regret guarantee of (FTRL-D), originally due to [33]:

**Theorem 1.** *Under* (FTRL-D)*, each player's regret is bounded as* $\text{Reg}_i(T) \leq H_i := \max h_i - \min h_i$.

Theorem 1 showcases the strong no-regret properties of (FTRL-D): it is not possible to guarantee less than constant, $\mathcal{O}(1)$ regret, so (FTRL-D) is optimal in this regard. In turn, by standard results [47], Theorem 1 implies further that the players' (correlated) empirical frequencies $z_{\alpha_1, \ldots, \alpha_N}(t) := (1/t) \int_0^t \prod_i x_{i\alpha_i}(\tau) \, d\tau$ converge to the game's set of coarse correlated equilibria (CCE) at a rate of $\mathcal{O}(1/t)$.

Importantly, this result makes no assumptions about the underlying game, but it does not carry the same predictive power in all games: for one thing, a game's set of CCE may include highly non-rationalizable outcomes (such as dominated strategies and the like) [61]; for another, the time-averaging that is inherent in the definition of empirical distributions may conceal a wide range of non-convergence phenomena, from cycles to chaos [48, 56]. On that account, the day-to-day behavior of (FTRL-D) in harmonic games cannot be understood from Theorem 1 alone, and requires a closer, more in-depth look.

Our first result below provides such a lense and shows that (FTRL-D) is almost-periodic in harmonic games, a property known as *Poincaré recurrence*.

**Theorem 2.** *Suppose* $\Gamma$ *is harmonic. Then almost every orbit* $x(t)$ *of* (FTRL-D) *returns arbitrarily close to its starting point infinitely often: specifically, for* (*Lebesgue*) *almost every initial condition* $x(0) = Q(y(0)) \in \mathcal{X}$*, there exists an increasing sequence of times* $t_n \uparrow \infty$ *such that* $x(t_n) \to x(0)$.

An immediate consequence of Theorem 2 is that no-regret learning under (FTRL-D) fails to converge in *any* harmonic game; in particular, since the orbits of (FTRL-D) eventually return to (almost) where they started, it is debatable if the players have learned anything at all, despite the fact that they incur at most constant regret. This cyclic, non-convergent landscape is the polar opposite of the long-run behavior of (FTRL-D) in *potential* games, where the dynamics are known to converge globally [24]. Thus, in addition to the strategic viewpoint of the previous section, Theorem 2 shows that harmonic games are orthogonal to potential games also from a *dynamic* viewpoint.

Theorem 2 also provides a far-reaching generalization of existing results on Poincaré recurrence in (possibly networked) two-player zero-sum games with an interior equilibrium [41] to general-sum, $N$-player games. Combined with our previous remark, and given that the zero-sum property is not as meaningful for $N$ players as it is for two,[7] the class of harmonic games can be seen as the more natural $N$-player generalization of two-player zero-sum games from a learning viewpoint.

To the best of our knowledge, the only comparable result to Theorem 2 in the literature is the very recent paper of Legacci et al. [35] who showed that the replicator dynamics – a special case of (FTRL-D) – are Poincaré recurrent in *uniform* harmonic games, that is, in harmonic games where the uniform distribution is a Nash equilibrium, cf. (A.1) and the discussion surrounding Definition 1. In this regard, Theorem 2 extends the recent results of Legacci et al. [35] along two axes: (*i*) it applies to the entire class of FTRL dynamics (not only the replicator dynamics); and (*ii*) it applies to the entire class of harmonic games (and not only *uniformly* harmonic games).

In terms of techniques, Legacci et al. [35] obtained their result through a surprising connection between a certain Riemannian metric underlying the replicator dynamics and the defining relation of uniformly harmonic games. This relation no longer holds for different instances of (FTRL-D) or for non-uniform harmonic games, so the techniques of [35] cannot be extended – and, in fact, Legacci et al. [35] stated this generalization as an open problem. Our techniques instead rely on the fact that the orbits $y(t)$ of (FTRL-D) comprise a volume-preserving flow in the game's payoff space $\mathcal{Y} \equiv \prod_i \mathcal{Y}_i$ (though not necessarily on $\mathcal{X}$), and then deriving a suitable constant of motion. In the case of the logit map (9), this constant of motion can be written as

$$G(x) = \prod_{i \in \mathcal{N}} \prod_{\alpha_i \in \mathcal{A}_i} x_{i\alpha_i}^{\mu_{i\alpha_i}} \qquad \text{for all } x \in \mathcal{X}, \tag{10}$$

where $\mu = (\mu_{i\alpha_i})_{\alpha_i \in \mathcal{A}_i, i \in \mathcal{N}}$ is the harmonic measure on $\mathcal{X}$ defining $\Gamma$. In the more general case, the construction of a constant of motion for (FTRL-D) involves a characterization of harmonic games in terms of a "strategic center", which we carry out in detail in Appendix C.

## 4  Discrete-time analysis: Convergence and constant regret via extrapolation

We now proceed to examine the regret and convergence properties of regularized learning algorithms in harmonic games. Starting with the standard, vanilla implementation of FTRL, we reproduce a well-known observation that FTRL spirals out to a non-terminating cycle of best-responses in Matching Pennies (which is a harmonic game). Subsequently, to correct this non-convergent behavior, we examine a flexible algorithmic template, which we call *extrapolated FTRL* (FTRL+), and which includes as special cases the optimistic and extra-gradient versions of FTRL.

---

[7]Recall that any $N$-player game can be turned into an equivalent zero-sum game by adding a fictitious player.

**4.1. Vanilla implementation of FTRL.** Building on the discussion of the previous section, the standard implementation of FTRL in discrete time for $n = 1, 2, \ldots$ is

$$x_{i,n+1} = \underset{p_i \in \mathcal{X}_i}{\arg\max}\left\{\sum_{k=1}^{n} u_i(p_i; x_{-i,n}) - \lambda_i h_i(p_i)\right\} = \underset{p_i \in \mathcal{X}_i}{\arg\max}\left\{\sum_{k=1}^{n} \langle v_i(x_k), p_i \rangle - \lambda_i h_i(p_i)\right\} \quad (11)$$

or, in more compact, iterative notation

$$y_{i,n+1} = y_{i,n} + \eta_i v_i(x_n) \qquad x_{i,n} = Q_i(y_{i,n}) \tag{FTRL}$$

where, as per (6), the map $Q_i \colon \mathcal{Y}_i \to \mathcal{X}_i$ denotes the *regularized choice map* of player $i \in \mathcal{N}$, $\lambda_i$ is a player-specific regularization weight parameter, and $\eta_i = 1/\lambda_i$ represents the *learning rate* of player $i$. Apart from their obvious differences – discrete vs. continuous time – a salient point that sets (FTRL) apart from (FTRL-D) is the inclusion of the parameter $\eta_i$; this parameter is necessary to control the algorithm's behavior, and we will discuss it in detail in the sequel.

As mentioned in the introduction, a major shortfall of (FTRL) – and one of the main reasons for the increased popularity of optimistic / extra-gradient methods – is that it may spiral away from Nash equilibrium, even in simple $2 \times 2$ games with a unique equilibrium. The standard example of this behavior is Matching Pennies, a two-player zero-sum game with a fully-mixed equilibrium which is also uniformly harmonic, so the trajectories of (FTRL-D) are Poincaré recurrent (and, in fact, periodic). In more detail, this game can be compactly represented by the payoff field $v(x_1, x_2) = (4x_2 - 2, 2 - 4x_1)$ for $x_1, x_2 \in [0, 1]$, and its unique Nash equilibrium is $x^* = (1/2, 1/2)$. Thus, if we run (FTRL) with a Euclidean regularizer – that is, $h_i(x_i) = x_i^2/2$ for $i = 1, 2$ – and the same learning rate $\eta$ for both players, a straightforward calculation shows that the distance $D_n = (x_{1,n} - x_1^*)^2/2 + (x_{2,n} - x_2^*)^2/2$ between $x_n$ and $x^*$ evolves as

$$D_{n+1} = \tfrac{1}{2}(x_{1,n} + \eta v_1(x_n) - x_1^*)^2 + \tfrac{1}{2}(x_{2,n} + \eta v_2(x_n) - x_2^*)^2 = (1 + 16\eta^2)D_n \tag{12}$$

as long as $x_n + \eta v(x_n) \in \mathcal{X}$. In other words, the distance of the iterates of (FTRL) from the game's equilibrium grows at a geometric rate until $x_n$ reaches the boundary of $\mathcal{X}$ and is ultimately trapped in a non-terminating cycle of best responses, cf. Fig. 1. In this regard, the rationality properties of (FTRL) are even worse than those of (FTRL-D) because the game's equilibrium is now *repelling*.

**4.2. Extrapolated FTRL.** To mitigate this undesirable, divergent behavior of (FTRL), a standard approach in the literature is the inclusion of a forward-looking, "*extrapolation step*". Instead of updating the algorithm's "base state" $x_n$ directly, players first move to an interim "leading state" $x_{n+1/2}$ using payoff information from $x_n$ (this is the extrapolation step); subsequently, players update $x_n$ using payoff information from the leading state $x_{n+1/2}$, and the process repeats. In this way, players attempt to anticipate their payoff landscape and, in so doing, to take a more informed update step at each iteration.

The seed of this idea goes back to Korpelevich [32] and Popov [49] in the context of solving monotone variational inequality problems, and it has since percolated to a wide array of "*extra-gradient*" or "*optimistic*" methods, such as the mirror-prox algorithm of Nemirovski [45], the dual extrapolation variant of Nesterov [46], the optimistic mirror descent algorithm of Chiang et al. [9] and Rakhlin & Sridharan [50], and many others. Given the different operational envelope of each of these methods, we consider below an integrated algorithmic template, which we call *extrapolated FTRL* (FTRL+), and which is sufficiently flexible to account for a broad range of these schemes.

Formally, the proposed algorithmic blueprint unfolds in two phases as follows:

a) *Extrapolation phase:* $\quad y_{i,n+1/2} = y_{i,n} + \eta_i \hat{v}_{i,n} \qquad x_{i,n+1/2} = Q_i(y_{i,n+1/2})$
b) *Update phase:* $\quad\qquad y_{i,n+1} = y_{i,n} + \eta_i \hat{v}_{i,n+1/2} \qquad x_{i,n+1} = Q_i(y_{i,n+1})$ $\qquad$ (FTRL+)

In the above, $\eta_i > 0$ is the learning rate of player $i$, $x_n$ and $x_{n+1/2}$ denote respectively the method's *base* and *leading* states at stage $n = 1, 2, \ldots$, and $\hat{v}_{i,n}$ and $\hat{v}_{i,n+1/2}$ are sequences of "black-box" payoff models at $x_n$ and $x_{n+1/2}$ respectively.

Specifically, following Azizian et al. [3], we will assume throughout that

$$\hat{v}_{i,n+1/2} = v_i(x_{n+1/2}) \quad \text{for all } i \in \mathcal{N} \text{ and all } n = 1, 2, \ldots \tag{13a}$$

i.e., players always update the base state $x_n$ using payoff information from the leading state $x_{n+1/2}$. By contrast, the leading state $x_{n+1/2}$ can be generated in many different ways, depending on the targeted update structure. In this regard, we will consider the linear model

$$\hat{v}_{i,n} = a_i\, v_i(x_n) + b_i\, v_i(x_{n-1/2}) \quad \text{for all } i \in \mathcal{N} \text{ and all } n = 1, 2, \ldots \tag{13b}$$

where the player-specific coefficients $a_i, b_i \geq 0$ satisfy $a_i + b_i \leq 1$ and represent a mix of past and present payoff information. In this way, depending on the values of $a_i$ and $b_i$, we obtain the following prototypical regularized learning methods as special cases of (FTRL+):

a) *FTRL:* if $a_i = b_i = 0$ for all $i \in \mathcal{N}$, players essentially forego any look-ahead efforts, so we get

$$\hat{v}_n = 0 \qquad \text{for all } n = 1, 2, \ldots \tag{14a}$$

In turn, this gives $x_{n+1/2} = x_n$, i.e., (FTRL+) regresses to (FTRL).

b) *Extra-Step FTRL:* if $a_i = 1$ and $b_i = 0$ for all $i \in \mathcal{N}$, we have

$$\hat{v}_n = v(x_n) \qquad \text{for all } n = 1, 2, \ldots \tag{14b}$$

i.e., players use payoff information from their current state to generate the leading state $x_{n+1/2}$. This update structure requires two payoff queries per iteration and its origins can be traced back to the work of Korpelevich [32]. Specifically, depending on the choice of $h_i$, it is essentially equivalent to the mirror-prox [45] and dual extrapolation [46] algorithms, it contains as a special case the forward-looking algorithm of [15, 42], etc.

c) *Optimistic FTRL:* if $a_i = 0$ and $b_i = 1$ for all $i \in \mathcal{N}$, we have

$$\hat{v}_n = v(x_{n-1/2}) \quad \text{for all } n = 1, 2, \ldots \tag{14c}$$

i.e., players reuse the latest available payoff information instead of making a fresh query at $x_n$ (so the algorithm only requires one payoff query per iteration). In this way, (FTRL+) recovers the optimistic algorithms of [9, 26, 50, 59], the OMW update scheme of [11, 59] when $Q = \Lambda$, etc.

Clearly, the list above is not exhaustive: many other configurations are possible, e.g., with different players using different parameter settings for $a_i$ and $b_i$, depending on the information they have at hand and any other individual considerations. To avoid needlessly complicating the analysis, our only standing assumption will be that $a_i + b_i > 0$ for all $i \in \mathcal{N}$ (since, otherwise, the benefits of the extrapolation step would vanish). In particular, by rescaling the players' learning rates if needed, we will normalize $a_i$ and $b_i$ to $a_i + b_i = 1$, leading to the convex model

$$\hat{v}_{i,n} = \lambda_i \, v_i(x_n) + (1 - \lambda_i) \, v_i(x_{n-1/2}) \tag{15}$$

for some arbitrarily chosen ensemble of player-specific *extrapolation coefficients* $\lambda_i \in [0, 1]$, $i \in \mathcal{N}$. *Remark.* To simplify the presentation of our results, we will assume throughout the rest of our paper that (FTRL+) is initialized with $y_1 = y_{1/2} = 0$.

**4.3. Analysis & results.** With all this in hand, we are finally in a position to state our main results for (FTRL+) in harmonic games. We begin by showing that (FTRL+) achieves order-optimal regret:

**Theorem 3.** *Suppose that each player in a harmonic game $\Gamma$ is following* (FTRL+) *with learning rate $\eta_i \leq m_i K_i [2(N + 2) \max_j m_j G_j]^{-1}$ and payoff models as per* (13a) *and* (15). *Then the individual regret of each player $i \in \mathcal{N}$ is bounded as*

$$\text{Reg}_i(T) := \max_{p_i \in \mathcal{X}_i} \sum_{n=1}^{T} [u_i(p_i; x_{-i,n}) - u_i(x_n)] \leq \frac{H_i}{\eta_i} + \frac{2G_i}{N + 2} \sum_{j \in \mathcal{N}} \frac{H_j}{\eta_j G_j} \tag{16}$$

*where $H_i = \max h_i - \min h_i$, and $G_i$ is the Lipschitz modulus of $v_i$.*

Even though Theorem 3 invites a natural comparison with the constant regret bound of Theorem 1, the continuous- and discrete-time settings are fundamentally different, so any conclusions drawn from such a comparison would be specious. Indeed, constant regret guarantees in the spirit of (16) are particularly rare in the context of discrete-time algorithms, and as far as we are aware, similar bounds have only been established for optimistic methods in variationally stable and two-player zero-sum games [27]; other than that – and always to the best of our knowledge – the tightest regret bounds available for general games (finite or convex) seem to be (poly)logarithmic [13, 14]. In this regard, just like the recurrence result of Theorem 2, the $\mathcal{O}(1)$ regret bound of Theorem 3 represents a significant extension of existing results on zero-sum games (and polylogarithmic regret in general games), and suggests that, from a learning viewpoint, harmonic games are the most natural generalization of two-player zero-sum games to a general $N$-player context. We defer the proof of Theorem 3 to Appendix D.

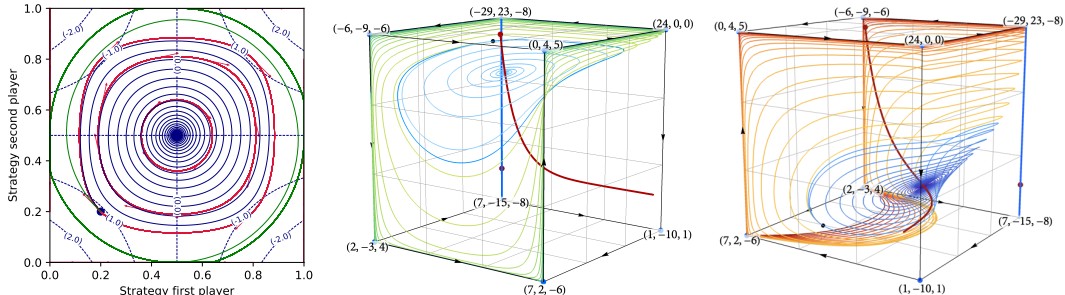

**Figure 1:** The evolution of vanilla vs. extrapolated FTRL schemes in harmonic games. In the left figure, we consider the game of Matching Pennies (blue: FTRL+; green: FTRL; red: continuous time FTRL); in the center and to the right, two different orbits in a $2 \times 2 \times 2$ harmonic game from two different viewpoints (blue: FTRL+; green/orange:FTRL; payoff profiles on vertices). In all cases, we ran the optimistic variant of FTRL+ ($\lambda_i = 0$ for all players), and we see that the trajectories of (FTRL) diverge away from equilibrium and the trajectories of (FTRL-D) are recurrent (actually, periodic), whereas (FTRL+) converges. We also see the highly non-convex structure of harmonic games as evidence by their equilibrium set (thick red line in center and right subfigures).

As an immediate corollary of the above, we conclude that, under (FTRL+), the empirical frequencies of play $z_{\alpha,n} := (1/n) \sum_{k=1}^n x_{\alpha,k}$, $\alpha \in \mathcal{A}$, converge to the game's set of CCE at a rate of $\mathcal{O}(1/n)$. This rate is, again, optimal, but as we discussed in Section 3, it offers little information in games where the marginalization of CCE does not lead to Nash equilibrium – and, in general $N$-player harmonic games, there is little hope that it would. In addition, even when the marginalization of CCE is Nash, the actual trajectory of play may – and, in fact, often *does* – behave very differently from the time-averaged frequency of play.

Despite these hurdles, we show below that (FTRL+) *does* converge to Nash equilibrium. To state this result formally, we will focus on the case where each player's regularizer is *smooth* in the sense that

$$h_i(x_i + t(x_i' - x_i)) \text{ is continuously differentiable at } t = 0 \tag{17}$$

for all $x_i \in \operatorname{im} Q_i$ and all $x_i' \in \mathcal{X}_i$.[8] Our prototypical examples – the entropic and Euclidean regularizers – both satisfy this mild requirement, as do all regularizers of the form $h_i(x_i) = \sum_{\alpha_i \in \mathcal{A}_i} \theta_i(x_{i\alpha_i})$ for some smooth convex function $\theta_i \colon [0, 1] \to \mathbb{R}$. We then have the following convergence result:

**Theorem 4.** *Suppose that each player in a harmonic game $\Gamma$ follows (FTRL+) with learning rate $\eta_i \leq m_i K_i [2(N + 2) \max_j m_j G_j]^{-1}$ and payoff models as per (13a) and (15). Then $x_n$ converges to the set of Nash equilibria of $\Gamma$.*

To the best of our knowledge, Theorem 4 is the first result of its kind for harmonic games – and, in that regard, it is somewhat unexpected. To be sure, two-player zero-sum games with a fully-mixed equilibrium exhibit a comparable pattern: FTRL is Poincaré recurrent in continuous time, its vanilla discretization is unstable, and its optimistic / forward-looking implementation is convergent. However, the convex-concave structure of min-max games which enables this analysis is completely absent in harmonic games, so it is less clear what to expect in this case (where even the set of Nash equilibria is non-convex, cf. Fig. 1). By this token, the convergence of (FTRL+) in harmonic games is a property that one could optimistically hope for, but not one that can be taken for granted.

From a technical standpoint, the proof of Theorems 3 and 4 involves two concurrent challenges:

1. Deriving a Lyapunov function with a "sufficient descent" property for all harmonic games.

2. Providing an integrated analysis for all possible update structures in (FTRL+).

With regard to the first point, our analysis hinges on the "energy function"

$$E(p, y) = \sum_{i \in \mathcal{N}} \frac{m_i}{\eta_i} F_i(p_i, y_i) \qquad p \in \mathcal{X}, y \in \mathcal{Y}, \tag{18}$$

In the above, $p \in \mathcal{X}$ is a benchmark strategy profile acting as a "reference point" for the analysis while

$$F_i(p_i, y_i) = \max_{x_i \in \mathcal{X}_i} \{\langle y_i, x_i \rangle - h_i(x_i)\} - [\langle y_i, p_i \rangle - h_i(p_i)] \tag{19}$$

---

[8]The restriction to $\operatorname{im} Q_i$ is technical in nature and is related to the subdifferentiability of $h_i$, cf. Appendix B.

denotes the *Fenchel coupling* associated to the regularizer $h_i$ of player $i \in \mathcal{N}$, and represents a "primal-dual" measure of divergence between $p_i \in \mathcal{X}_i$ and $y_i \in \mathcal{Y}_i$ (for an in-depth discussion, see Appendices B and D). Then, letting $E_n = E(p, y_n)$, the heavy lifting for our analysis is provided by the "template inequality"

$$
\begin{aligned}
E_{n+1} \leq E_n &+ \sum_{i \in \mathcal{N}} m_i \langle v_i(x_{n+1/2}), x_{i,n+1/2} - p_i \rangle \\
&+ \sum_{i \in \mathcal{N}} m_i \langle v_i(x_{n+1/2}) - v_i(x_n), x_{i,n+1} - x_{i,n+1/2} \rangle \\
&+ \sum_{i \in \mathcal{N}} m_i(1 - \lambda_i)\langle v_i(x_n) - v_i(x_{n-1/2}), x_{i,n+1} - x_{i,n+1/2} \rangle \\
&- \sum_{i \in \mathcal{N}} \frac{m_i K_i}{\eta_i}\left[\|x_{i,n+1} - x_{i,n+1/2}\|^2 + \|x_{i,n+1/2} - x_{i,n}\|^2\right].
\end{aligned}
\tag{20}
$$

A first important consequence of (20) is that the sequences $A_n = \|x_{n+1} - x_{n+1/2}\|^2$ and $B_n = \|x_{n+1/2} - x_n\|^2$ are both summable: this requires a repeated use of the Fenchel-Young inequality, and an instantiation of $p$ to the strategic center $q$ of $\Gamma$; we detail the relevant arguments in Appendices A and D. Then, by establishing a similar template inequality for *each* player $i \in \mathcal{N}$, we are able to bound the players' individual regret by the same upper bound that we derived for $\sum_n A_n$ and $\sum_n B_n$, and which is (up to certain secondary factors) the bound (16).

For the convergence to Nash equilibrium, the summability argument above also plays a crucial role. First, by a standard result on numerical sequences, the summability of $A_n$ and $B_n$ coupled with the template inequality (20) implies that the energy $E_n$ of the algorithm relative to the game's strategic center converges to some limit value $E_\infty$. In turn, this implies that the score sequence $y_n$ is bounded up to a multiple of the vector $(1, \ldots, 1)$, which corresponds to a constant payoff shift in the underlying game. Then, by focusing on convergent subsequences of $y_n$ and the optimality condition resulting from the definition of $Q$, we are able to show that any limit point of $v(x_n)$ satisfies the variational characterization (VI) of Nash equilibria, from which our claim follows.

## 5 Concluding remarks

Our results suggest that the long-run behavior of no-regret algorithms and dynamics in harmonic games is a very rich topic, and one which opens the door to an entirely new class of games where positive convergence results can be obtained. We find this particularly appealing, not only because harmonic games comprise the strategic complement of potential games, but also because they go beyond standard problems with a convex structure – for instance, even their equilibrium set is non-convex. As such, the fact that it is possible to obtain optimal regret guarantees and positive equilibrium convergence results in this setting is very promising for future work on the topic.

In terms of open questions, it would be important to examine the rate of convergence of (FTRL+) to equilibrium. Even though (FTRL+) has order-optimal regret bounds, this only helps in establishing a convergence rate to the game's set of coarse correlated equilibria; for Nash equilibria, earlier work by Golowich et al. [19] and some more recent results by Cai et al. [5] and Gorbunov et al. [20] have shed some light on the convergence of constrained Euclidean optimistic methods, but the technology therein does not extend to non-monotone, non-Euclidean problems. Inspired by Wei et al. [62], we conjecture that the convergence rate of (FTRL+) in harmonic games is linear: this is based on the observation that any harmonic game admits a fully-mixed Nash equilibrium, and the weighted sum in the definition of a harmonic game looks formally similar to the condition needed to establish metric subregularity in [62]; however, a proof would likely require different techniques.

Another important research direction has to do with the information available to the players. A first open question here concerns the case where players do not have access to full information on their mixed payoff vectors, but can only observe their pure payoffs – either in a "what if", counterfactual manner, or in the form of bandit, payoff-based feedback. In a similar manner, the algorithms presented here are not adaptive, in the sense that the players' step-size policy has to satisfy a certain bound that depends on correctly estimating some of the game's parameters. Obtaining an adaptive version of (FTRL+) which, in the spirit of Rakhlin & Sridharan [50] and Hsieh et al. [27, 28, 29], remains convergent and attains order-optimal regret in both adversarial and game-theoretic settings without any pre-play tuning is also an ambitious question for future research.

## Acknowledgments and Disclosure of Funding

This research was supported in part by the French National Research Agency (ANR) in the framework of the PEPR IA FOUNDRY project (ANR-23-PEIA-0003), the "Investissements d'avenir program" (ANR-15-IDEX-02), the LabEx PERSYVAL (ANR-11-LABX-0025-01), MIAI@Grenoble Alpes (ANR-19-P3IA-0003), the project IRGA2024-SPICE-G7H-IRG24E90, and NSF grant CCF2212233. PM is also with the Archimedes Research Unit/Athena RC & NKUA and acknowledges financial support by project MIS 5154714 of the National Recovery and Resilience Plan Greece 2.0 funded by the European Union under the NextGenerationEU Program.

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

# Appendices

## A  Harmonic Games

The class of *uniform harmonic games* (UHGs) introduced by Candogan et al. [6] provides a game-theoretic framework for modeling strategic situations with conflicting, anti-aligned interests.[9] Broadly speaking, the characterizing property of uniform harmonic games is the following: for any player considering a deviation towards a specific pure strategy profile, there exist other players who are motivated to deviate *away* from that profile.

Given a finite game $\Gamma = \Gamma(\mathcal{N}, \mathcal{A}, u)$, this is formalized by the condition that, for all $\alpha \in \mathcal{A}$,

$$\sum_{i \in \mathcal{N}} \sum_{\beta_i \in \mathcal{A}_i} \left[ u_i(\alpha_i; \alpha_{-i}) - u_i(\beta_i; \alpha_{-i}) \right] = 0 \,. \tag{A.1}$$

From a strategic viewpoint, uniform harmonic games complement potential games: Candogan et al. [6] showed that any finite game can be uniquely decomposed into the sum of a potential game and a uniform harmonic game, up to linear transformations of the payoff functions that do not change the strategic structure of the game.

Since their introduction, harmonic games have generated a substantial body of literature; for a brief survey, we refer the reader to [35].

**A.1.  Harmonic games, measures and comeasures.** The class of uniform harmonic games exhibits intriguing, yet restrictive, properties. Notably, a UHG always admits the uniformly mixed strategy as

---

[9]We include here the word "uniform" to distinguish the class of harmonic games introduced by Candogan et al. [6] from the more general class of harmonic games considered in this work, cf. Definition 1.

a NE, and it generally possesses a *continuum* of Nash equilibria [6]. Additionally, the framework of UHGs and the decomposition proposed by Candogan et al. [6] are incompatible with common game-theoretical transformations, such as the duplication of strategies or rescaling of payoffs [1]. To address the above limitations, Abdou et al. [1] extended the definition of harmonic games by the introduction of two parameters: a *measure*, that is a positive weight each player assigns to each of *their own* strategy; and a *comeasure*, that is a positive weight each player assigns to each of the *other players'* action profiles.

**Definition A.1.** Let $\Gamma(\mathcal{N}, \mathcal{A}, u)$ be a finite game. A *player measure* $\mu_i$ is a function $\mu_i \colon \mathcal{A}_i \to \mathbb{R}_{++}$; a *player co-measure* $\gamma_i$ is a function $\gamma_i \colon \mathcal{A}_{-i} \to \mathbb{R}_{++}$. Correspondingly, a collection $\mu = \{\mu_i\}_{i \in \mathcal{N}}$ (resp. $\gamma = \{\gamma_i\}_{i \in \mathcal{N}}$) of player measures (resp. comeasures) is called *game measure* (resp. *game comeasure*). If $\mu_i$ is a player measure, we will write $|\mu_i| := \sum_{\alpha_i} \mu_{i\alpha_i}$. Finally, a *probability measure* is a game measure $\mu$ such that $|\mu_i| = 1$ for all $i \in \mathcal{N}$; a *uniform measure* is a game measure $\mu$ such that $\mu_{i\alpha_i} = 1$ for all $i \in \mathcal{N}, \alpha_i \in \mathcal{A}_i$; and a *uniform comeasure* is a game comeasure $\gamma$ such that $\gamma_{i\alpha_{-i}} = 1$ for all $i \in \mathcal{N}, \alpha_{-i} \in \mathcal{A}_{-i}$.

With these notions in place, Abdou et al. [1] define a finite game $\Gamma$ to be $(\mu, \gamma)$-*harmonic* if there exist a game measure $\mu$ and a game comeasure $\gamma$ such that, for all $\alpha \in \mathcal{A}$,

$$\sum_i \sum_{\beta_i} \mu_{i\beta_i} \gamma_{i\alpha_{-i}} \left[ u_i(\alpha_i; \alpha_{-i}) - u_i(\beta_i; \alpha_{-i}) \right] = 0 \,. \tag{A.2}$$

In this work, we focus solely on harmonic games with *uniform comeasure*. As discussed after Definition 1 in the main body of the article, this choice comes without loss of generality: the game comeasure in Eq. (A.2) can be absorbed by a payoff rescaling to give a game that is still harmonic, and *preference equivalent* to the original game – in a sense that we make precise in the next section.

**A.2. Preference equivalence between harmonic games.** The strategic structure of a game is preserved under monotonic transformations of the utility functions, since the set of pure Nash equilibria of a game is an ordinal object – it depends only on the signs of unilateral payoff differences, and not on their absolute values. For this reason, two games $\Gamma(\mathcal{N}, \mathcal{A}, u)$ and $\Gamma'(\mathcal{N}, \mathcal{A}, u')$ are called *preference-equivalent* (PE) if for all $\alpha, \beta \in \mathcal{A}$ and all $i \in \mathcal{N}$, we have

$$\text{sgn} \left[ u'_i(\beta_i; \alpha_{-i}) - u'_i(\alpha_i; \alpha_{-i}) \right] = \text{sgn} \left[ u_i(\beta_i; \alpha_{-i}) - u_i(\alpha_i; \alpha_{-i}) \right]. \tag{A.3}$$

Two games are *strategically equivalent* (SE) – and we write $\Gamma \sim \Gamma'$ – if they have the same unilateral payoff differences, that is if

$$u'_i(\beta_i; \alpha_{-i}) - u'_i(\alpha_i; \alpha_{-i}) = u_i(\beta_i; \alpha_{-i}) - u_i(\alpha_i; \alpha_{-i}) \tag{A.4}$$

for all $\alpha, \beta \in \mathcal{A}$ and all $i \in \mathcal{N}$; strategically equivalent games are clearly preference-equivalent.

**Lemma A.2.** *Let* $\Gamma_{\mu,\gamma} = \Gamma_{\mu,\gamma}(\mathcal{N}, \mathcal{A}, u)$ *be a harmonic game in the sense of Eq. (A.2). Then the game* $(\mathcal{N}, \mathcal{A}, u')$ *with* $u'_i(\alpha_i; \alpha_{-i}) = \gamma_{i\alpha_{-i}} u_i(\alpha_i; \alpha_{-i})$ *is preference-equivalent to the game* $\Gamma_{\mu,\gamma}$, *and it is harmonic in the sense of Eq. (A.2) with measure* $\mu$ *and uniform comeasure.*

*Proof.* Let $u''_i(\alpha_i; \alpha_{-i}) = \mu_{i\alpha_i} \gamma_{i\alpha_{-i}} u_i(\alpha_i; \alpha_{-i})$. Then replacing above, for all $\alpha \in \mathcal{A}$,

$$0 = \sum_{i \in \mathcal{N}} \sum_{\beta_i \in \mathcal{A}_i} \mu_{i\beta_i} \left[ \frac{u''_i(\alpha_i; \alpha_{-i})}{\mu_{i\alpha_i}} - \frac{u''_i(\beta_i; \alpha_{-i})}{\mu_{i\beta_i}} \right] \,.$$

Let $u'_i(\alpha_i; \alpha_{-i}) = \frac{u''_i(\alpha_i; \alpha_{-i})}{\mu_{i\alpha_i}} = \gamma_{i\alpha_{-i}} u_i(\alpha_i; \alpha_{-i})$. The game $u'$ is preference-equivalent to $u$, and

$$0 = \sum_{i \in \mathcal{N}} \sum_{\beta_i \in \mathcal{A}_i} \mu_{i\beta_i} \left[ u'_i(\alpha_i; \alpha_{-i}) - u'_i(\beta_i; \alpha_{-i}) \right] \tag{A.5}$$

for all $\alpha \in \mathcal{A}$, so $u'$ is harmonic in the sense of A.2 with measure $\mu$ and uniform comeasure. ∎

In the proof above we perform the intermediate step $u \to u''$ rather than defining directly $u \to u'$ to stress the difference between rescaling the payoffs of a game by a game measure $\mu$ and by a game comeasure $\gamma$. The game with payoffs $u' = \gamma u$ (the meaning of this notation made precise in the proof above) is preference-equivalent to the game with payoffs $u$, i.e., rescaling the payoffs by a comeasure does not change the strategic structure of the game. On the other hand, the game with

payoffs $u'' = \mu u'$ – again, the meaning made precise in the proof – is *not* PE to the game with payoffs $u'$: rescaling the payoffs by a measure can change the preferences of the players, and leads to a game with intrinsically different strategic structure.

Lemma A.2 motivates our choice to focus in this work on harmonic games with arbitrary measures and uniform comeasures, and to adopt (HG) from Definition 1 to characterize harmonic games: a *harmonic game* (HG) $\Gamma_\mu = \Gamma_\mu(\mathcal{N}, \mathcal{A}, u)$ is a finite game $(\mathcal{N}, \mathcal{A}, u)$ with a game measure $\mu$ such that (HG) holds, i.e., $\sum_{i \in \mathcal{N}} \sum_{\beta_i \in \mathcal{A}_i} \mu_{i\beta_i}[u_i(\alpha_i; \alpha_{-i}) - u_i(\beta_i; \alpha_{-i})] = 0$ for all $\alpha \in \mathcal{A}$.

**A.3. Mixed characterization of harmonic games.** The defining property (HG) allows for an equivalent characterization of harmonic games in terms of their mixed payoffs:

**Lemma A.3.** *A finite game $\Gamma = \Gamma(\mathcal{N}, \mathcal{A}, u)$ is harmonic with measure $\mu$ if and only if*

$$\sum_{i \in \mathcal{N}} |\mu_i| \left\langle v_i(x), x_i - \frac{\mu_i}{|\mu_i|} \right\rangle = 0 \quad \text{for all } x \in \mathcal{X}. \qquad \text{(HG-mixed)}$$

*Proof.* Given a finite game $\Gamma = \Gamma(\mathcal{N}, \mathcal{A}, u)$ and a game measure $\mu$, let $F_i : \mathcal{A} \to \mathbb{R}$ be defined by $F_i(\alpha) = \sum_{\beta_i \in \mathcal{A}_i} \mu_{i\beta_i}[u_i(\alpha_i; \alpha_{-i}) - u_i(\beta_i; \alpha_{-i})]$. By definition, $\Gamma$ is a $\mu$-harmonic game if and only if $F(\alpha) := \sum_{i \in \mathcal{N}} F_i(\alpha) = 0$ for all $\alpha \in \mathcal{A}$. Denote (with slight abuse of notation) by $F : \mathcal{X} \to \mathbb{R}$ the multilinear extension of $F : \mathcal{A} \to \mathbb{R}$, i.e., $F(x) = \sum_\alpha x_\alpha F(\alpha)$, with $x_\alpha := \prod_i x_{i\alpha_i}$. Now, $F(\alpha) = 0$ for all $\alpha \in \mathcal{A}$ if and only if $F(x) = 0$ for all $x \in \mathcal{X}$, which is the case if and only if

$$0 = F(x) = \sum_\alpha x_\alpha \sum_i F_i(\alpha) = \sum_i \sum_{\alpha_i} \sum_{\alpha_{-i}} x_{i\alpha_i} x_{-i\alpha_{-i}} \sum_{\beta_i} \mu_{i\beta_i}[u_i(\alpha_i; \alpha_{-i}) - u_i(\beta_i; \alpha_{-i})]$$

$$= \sum_i \sum_{\beta_i} \mu_{i\beta_i}[u_i(x_i; x_{-i}) - u_i(\beta_i; x_{-i})] = \sum_i \left[|\mu_i|\langle v_i(x), x_i \rangle - \langle v_i(x), \mu_i \rangle\right] \quad \text{for all } x \in \mathcal{X},$$

from which we conclude by factoring out the $|\mu_i|$ terms. ∎

*Remark.* The first equality in the second line holds true for harmonic games with *uniform* comeasure $\gamma_{i\alpha_{-i}} = 1$, since $\gamma_{i\alpha_{-i}} \neq 1$ terms would couple with the corresponding $x_{-i\alpha_{-i}}$ terms in the sum.

The above result can be reformulated as follows:

**Proposition A.4.** *A finite game $\Gamma = \Gamma(\mathcal{N}, \mathcal{A}, u)$ is harmonic if and only if it admits a* strategic center *$(m, q)$, viz. if there exist (i) a vector $m \in \mathbb{R}_{++}^N$ and (ii) a fully mixed strategy $q \in \mathcal{X}$ such that*

$$\sum_{i \in \mathcal{N}} m_i \langle v_i(x), x_i - q_i \rangle = 0 \quad \text{for all } x \in \mathcal{X}. \qquad \text{(HG-center)}$$

This expression is intriguing: it suggest that a game is harmonic precisely if there exists a fully mixed strategy $q$ such that, for all $x \in \mathcal{X}$, the payoff vector $v(x)$ is perpendicular (with respect to a $m$-weighted inner product) to $x - q$; cf. Example A.1 and Fig. 2. The striking dynamical consequences of this "circular" strategic structure – hinted at in Fig. 2, showing a *periodic* orbit of FTRL in continuous time – are captured precisely by Theorem 2 in the main text.

*Proof of Proposition A.4.* Let $\Gamma_\mu = \Gamma_\mu(\mathcal{N}, \mathcal{A}, u)$ be harmonic; then by Lemma A.3 that there exist a strategic center $(m, q)$ given by $m_i := |\mu_i|$ and $q_i := \mu_i/|\mu_i|$ with $i \in \mathcal{N}$. Conversely let $\Gamma = \Gamma(\mathcal{N}, \mathcal{A}, u)$ admit a strategic center $(m, q)$; then by the same argument $\Gamma$ is harmonic with $\mu_i := m_i q_i$ for all $i \in \mathcal{N}$. ∎

An immediate corollary is the following:

**Corollary A.5.** *If a finite game $\Gamma$ admits a strategic center $(m, q)$, then $q$ is a Nash equilibrium.*

*Proof.* By Proposition A.4 if $\Gamma$ admits a strategic center $(m, q)$ then it is $\mu$-harmonic with $\mu_i = m_i q_i$ for all $i \in \mathcal{N}$; and $(\mu_i/|\mu_i|)_{i \in \mathcal{N}}$ is always a NE for $\mu$-harmonic games [1, Theorem 1]. ∎

*Remark.* The converse does not hold: a fully mixed Nash equilibrium is not necessarily a strategic center. If it were, a game would be harmonic precisely if it admitted a fully mixed NE, which is not the case – think for example of coordination or anti-coordination games, that admit a fully mixed Nash equilibrium and are not harmonic.

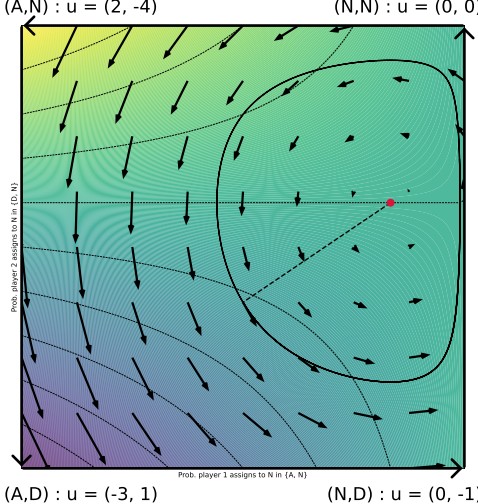

(A,N) : u = (2, -4)          (N,N) : u = (0, 0)

Prob. player 2 assigns to N in (D, N)

Prob. player 1 assigns to N in (A, N)

(A,D) : u = (-3, 1)          (N,D) : u = (0, -1)

**Figure 2:** Representation of the harmonic payoff structure for the game in Example A.1. Each payoff vector $v(x)$ (black arrows) is perpendicular (with respect to a weighted inner product) to the vector $x - q$ (dotted segment) between the evaluation point $x$ of the payoff field and the fully mixed Nash equilibrium $q$ (red point). As a consequence every orbit of FTRL in continuous time (such as the one represented by the black curve) is *Poincaré recurrent* (in this low-dimensional example, even periodic), as detailed in Theorem 2 in the main text. Color shading and dotted lines represents player 1's utility level sets, with brighter regions indicating higher payoffs.

**Example A.1** (A harmonic game: Siege)**.** Consider the following $2 \times 2$ game: an army (the row player) must choose between Attacking a fortress (pure strategy $A$ ) and Not attacking (pure strategy $N$ ). Simultaneously, the fortress (the column player) decides whether to activate its Defenses (pure strategy $D$ ) or Not (pure strategy $N$ ). Engaging in either action (the attack or the defense) incurs a preparation cost of $c > 0$. The army gains $a_s > c$ if it attacks an undefended fortress, but suffers a loss of $a_f > 0$ if it attacks and encounters defenses (the subscripts $s$ and $f$ standing respectively for "successful" and "failed"). Conversely, the fortress benefits by $d_s > 0$ if it is defended against an attack, while it incurs a loss of $d_f > 0$ if attacked without defenses; defeating the attacking army is worth the preparation cost for the fortress, namely $d_s - c > -d_f$. This scenario is captured by the following payoff matrix, specialized on the right to the case $c = 1, a_s = 3, a_f = 2, d_s = 2, d_f = 4$:

$$
\begin{array}{c|cc}
 & D & N \\
\hline
A & (-a_f - c,\ d_s - c) & (a_s - c,\ -d_f) \\
N & (0,\ -c) & (0,\ 0)
\end{array}
\qquad
\begin{array}{c|cc}
 & D & N \\
\hline
A & -3, 1 & 2, -4 \\
N & 0, -1 & 0, 0
\end{array}
\tag{A.6}
$$

To determine if the game is harmonic, look for a solution of the linear system

$$
\sum_{i \in \mathcal{N}} \sum_{\beta_i \in \mathcal{A}_i} \mu_{i\beta_i} [u_i(\alpha_i; \alpha_{-i}) - u_i(\beta_i; \alpha_{-i})] = 0 \quad \text{for all } \alpha \in \mathcal{A}, \tag{HG}
$$

subject to the constraints $\mu_{i\alpha_i} > 0$ for all $i \in \mathcal{N}, \alpha_i \in \mathcal{A}_i$. For a fixed payoff function $u$, this is a system of $\prod_{j \in \mathcal{N}} A_j$ linear equations (one for each $\alpha \in \mathcal{A}$) in the $\sum_{j \in \mathcal{N}} A_j$ variables $((\mu_{i\alpha_i})_{\alpha_i \in \mathcal{A}_i})_{i \in \mathcal{N}}$, where $A_i$ is the number of pure actions of player $i \in \mathcal{N}$. With $u$ given by (A.6) – left,

$$
\mu = \lambda \left[ \left( \frac{c}{a_f + c}, \frac{-c + d_f + d_s}{a_f + c} \right), \left( \frac{a_s - c}{a_f + c}, 1 \right) \right] \tag{A.7}
$$

is a feasible solution of (HG) for any $\lambda > 0$, so the game is harmonic with a 1-dimensional set of measures. The corresponding strategic center $(m, q)$ with $m_i = \sum_{\alpha_i} \mu_{i\alpha_i}, q_i = \mu_i/m_i, i \in \{1, 2\}$ is

$$
m = \lambda \left( \frac{d_f + d_s}{a_f + c}, \frac{a_f + a_s}{a_f + c} \right), \quad q = \left[ \left( \frac{c}{d_f + d_s}, \frac{-c + d_f + d_s}{d_f + d_s} \right), \left( \frac{a_s - c}{a_f + a_s}, \frac{a_f + c}{a_f + a_s} \right) \right]. \tag{A.8}
$$

As a sanity check, compute the payoff field and verify that (HG-center) holds true in the specialized case (A.6) – right. Denoting the mixed strategies of players 1 and 2 respectively by $x \in \Delta(\{A, N\})$ and $y \in \Delta(\{D, N\})$, the payoff fields are $v_1(x, y) = (-3y_D + 2y_N, 0), \quad v_2(x, y) =$

$(x_A - x_N, -4x_A)$. Choosing $\lambda = 3$ the strategic center gives weights $m = (6, 5)$ and Nash equilibrium $q = [(1/6, 5/6), (2/5, 3/5)]$. Condition (HG-center) boils down to $6 \langle v_1, x - q_1 \rangle + 5 \langle v_2, y - q_2 \rangle = 0$, which one readily verifies to hold true by replacing the expressions above and recalling that $x_A + x_N = 1 = y_D + y_N$. Fig. 2 illustrates the situation: each payoff vector $v(x)$ (black arrows) is perpendicular (with respect to a weighted inner product) to the vector $x - q$ (dotted segment) between the evaluation point $x$ of the payoff field and the fully mixed Nash equilibrium $q$ (red point). ◇

**A.4. Harmonic and zero-sum games.** Candogan et al. [6]'s uniform harmonic games, defined by Eq. (A.1), are precisely the harmonic games with uniform measure, which makes uniform harmonic games a strict subset of the set of HGs. Importantly, HGs include another archetypal class of perfect-competition games: as we show in this section, *two-player zero-sum games* (2ZSGs) with an interior NE $x^*$ are harmonic with (probability) measure $\mu = x^*$.

To show this, we will need the following definition and lemma:

**Definition A.6** (Non-strategic game). A finite normal form game $\Gamma = \Gamma(\mathcal{N}, \mathcal{A}, k)$ is called *non-strategic* if the payoff of each player does not depend on their own choice, viz. if $k_i(\alpha_i, \alpha_{-i}) = k_i(\beta_i, \alpha_{-i})$ for all $i \in \mathcal{N}, \alpha \in \mathcal{A}, \beta_i \in \mathcal{A}_i$.

**Lemma A.7.** *Two finite games $\Gamma(\mathcal{N}, \mathcal{A}, u), \Gamma'(\mathcal{N}, \mathcal{A}, u')$ are strategically equivalent in the sense of Eq. (A.4) if and only if their difference is a non-strategic game.*

*Proof.* Let $\Gamma - \Gamma'$ be non-strategic; then $k := u' - u$ fulfills the condition of Definition A.6, which shows that $u$ and $u'$ fulfill Eq. (A.4). Conversely let $\Gamma$ and $\Gamma'$ be strategically equivalent; set $k := u' - u$ and rearrange the terms in Eq. (A.4) to immediately conclude that $k$ is a non-strategic game. ∎

**Proposition A.8.** *Let $\Gamma_\mu = \Gamma_\mu(\mathcal{N}, \mathcal{A}, u)$ be a harmonic game. If the measure $\mu$ fulfills $|\mu_i| = |\mu_j|$ for all $i, j \in \mathcal{N}$ then $\Gamma_\mu$ is strategically equivalent to a zero-sum game.*

*Proof.* Recall that $|\mu_i| \equiv \sum_{\alpha_i} \mu_{i\alpha_i}$. Under the assumption $|\mu_i| = |\mu_j|$ for all $i, j \in \mathcal{N}$, let $c := |\mu_i|$ for any $i \in \mathcal{N}$. By (HG), the payoff $u$ of $\Gamma_\mu$ in this case fulfills $\sum_{i \in \mathcal{N}}[u_i(\alpha) - k_i(\alpha)] = 0$ for all $\alpha \in \mathcal{A}$, with $k_i(\alpha_i; \alpha_{-i}) := c^{-1} \sum_{\beta_i} \mu_{i\beta_i} u_i(\beta_i, \alpha_{-i})$. Set $u'_i := u_i - k_i$. By definition $u'$ is a zero-sum game; furthermore, the difference between $u_i$ and $u'_i$ is non-strategic, since $k_i(\alpha_i; \alpha_{-i})$ does not depend on $\alpha_i$. Thus $u_i$ and $u'_i$ are strategically equivalent by Lemma A.7. ∎

In particular we have the following:

**Corollary A.9.** *Let $\Gamma_\mu = \Gamma_\mu(\mathcal{N}, \mathcal{A}, u)$ be a harmonic game. If the measure $\mu$ is a probability measure, then $\Gamma_\mu$ is strategically equivalent to a zero-sum game.*

The converse holds true only in the case of two-player games:

**Proposition A.10.** *Every two-player zero-sum game with an interior Nash equilibrium $x^*$ is harmonic, with (probability) measure $\mu = x^*$.*

*Proof.* Let $\Gamma = \Gamma(\mathcal{N}, \mathcal{A}, u)$ be a two-player zero-sum game with interior Nash equilibrium $x^*$. If we show that

$$\sum_{i \in \mathcal{N}} |x_i^*| \left\langle v_i(x), x_i - \frac{x_i^*}{|x_i^*|} \right\rangle = 0 \quad \text{for all } x \in \mathcal{X}, \tag{A.9}$$

then we can conclude by Lemma A.3 that $\Gamma$ is harmonic with measure $x^*$. Eq. (A.9) holds indeed true: $|x_i^*| = 1$ for all $i \in \mathcal{N}$, and it is well known [41, 42] that two-player zero-sum games with an interior equilibrium $x^*$ fulfill $\sum_{i \in \mathcal{N}} \langle v_i(x), x_i - x_i^* \rangle = 0$ for all $x \in \mathcal{X}$, so we are done. ∎

Harmonic games thus encompass and substantially generalize two prototypical classes of games with anti-aligned incentives, serving as an ideal complement to the class of potential games. This is made precise in [1]: building on the work of Candogan et al. [6], Abdou et al. [1] showed that, for any choice of game measure $\mu$, every finite game can be uniquely decomposed into the sum of a potential and a $\mu$-harmonic game, up to strategic equivalence.

This establishes harmonic games as the natural complement of potential games from a strategic perspective; Theorem 2 in the main text shows that this holds true from a *dynamic* perspective as well.

# B  Basic properties of regularizers and the induced choice maps

In this appendix, we collect a number of properties concerning regularizers and the associated choice maps. To avoid carrying around the player index $i \in \mathcal{N}$, we state all our results for a generic convex subset $\mathcal{C}$ of some real vector space $\mathcal{V}$. The desired properties for FTRL will then be obtained by specializing $\mathcal{C}$ to $\mathcal{X}_i$ or $\mathcal{X}$ and $\mathcal{V}$ to $\mathbb{R}^{\mathcal{A}_i}$ or $\prod_j \mathbb{R}^{\mathcal{A}_j}$, depending on the context.

**B.1. Preliminary definitions.** To begin, let $\mathcal{V}$ be a $d$-dimensional normed space with norm $\|\cdot\|$. In what follows, we will write $\mathcal{Y} := \mathcal{V}^*$ for the dual space of $\mathcal{V}$, $\langle y, x \rangle$ for the canonical pairing between $x \in \mathcal{V}$ and $y \in \mathcal{V}^*$, and $\|y\|_* = \max\{\langle y, x \rangle : \|x\| \leq 1\}$ for the induced dual norm on $\mathcal{Y}$. Following standard conventions in convex analysis, functions will be allowed to take values in the extended real line $\mathbb{R} \cup \{\infty\}$, and if $f : \mathcal{V} \to \mathbb{R} \cup \{\infty\}$ is a convex function on $\mathcal{V}$, we will denote its *effective domain* as

$$\operatorname{dom} f := \{x \in \mathcal{V} : f(x) < \infty\} . \tag{B.1}$$

In addition, assuming $\operatorname{dom} f \neq \varnothing$, the *subdifferential* of $f$ at $x$ is defined as

$$\partial f(x) := \{y \in \mathcal{Y} : f(x') \geq f(x) + \langle y, x' - x \rangle \text{ for all } x' \in \mathcal{V}\} \tag{B.2}$$

and we denote the *domain of subdifferentiability* of $f$ as

$$\operatorname{dom} \partial f = \{x \in \mathcal{V} : \partial f(x) \neq \varnothing\} . \tag{B.3}$$

Finally, to ease notation, a convex function $f : \mathcal{C} \to \mathbb{R}$ will be identified with the extended-real-valued function $\bar{f} : \mathcal{V} \to \mathbb{R} \cup \{\infty\}$ that agrees with $f$ on $\mathcal{C}$ and is identically equal to $\infty$ on $\mathcal{V} \setminus \mathcal{C}$.

With all this in hand, let $\mathcal{C}$ be a closed convex subset of $\mathcal{V}$, and let $h : \mathcal{C} \to \mathbb{R}$ be a $K$-strongly convex *regularizer* on $\mathcal{C}$, that is,

$$h(tx + (1-t)x') \leq th(x) + (1-t)h(x') - \frac{K}{2}t(1-t)\|x' - x\|^2 . \tag{B.4}$$

By standard arguments in convex analysis, this readily implies that

$$h(x') \geq h(x) + \partial h(x; x' - x) + \frac{K}{2}\|x' - x\|^2 \quad \text{for all } x, x' \in \mathcal{X}, \tag{B.5}$$

where

$$\partial h(x; x' - x) = \lim_{\theta \to 0^+} [h(x + \theta(x' - x)) - h(x)]/\theta \tag{B.6}$$

denotes the one-sided directional derivative of $h$ at $x$ along the direction of $x' - x$. To proceed, we will need the following basic objects:

1. The *convex conjugate* $h^* : \mathcal{Y} \to \mathbb{R}$ of $h$:

$$h^*(y) = \max_{x \in \mathcal{X}}\{\langle y, x \rangle - h(x)\} \qquad \text{for all } y \in \mathcal{Y}. \tag{B.7}$$

2. The *regularized choice map* – or *mirror map* – $Q : \mathcal{Y} \to \mathcal{X}$ induced by $h$:

$$Q(y) = \arg\max_{x \in \mathcal{X}}\{\langle y, x \rangle - h(x)\} \qquad \text{for all } y \in \mathcal{Y} \tag{B.8}$$

3. The associated *Fenchel coupling* $F : \mathcal{X} \times \mathcal{Y} \to \mathbb{R}$ of $h$:

$$F(p, y) = h(p) + h^*(y) - \langle y, p \rangle \qquad \text{for all } p \in \mathcal{X}, y \in \mathcal{Y}. \tag{B.9}$$

*Remark.* The terminology "Fenchel coupling" is due to [38, 40], which we follow closely in terms of notation and conventions.

The proposition below provides some basic properties concerning the first two objects above:

**Proposition B.1.** *Let $h$ be a $K$-strongly convex regularizer on $\mathcal{C}$. Then:*

(a) *$Q$ is single-valued on $\mathcal{Y}$; in particular, for all $x \in \operatorname{dom} \partial h$ and all $y \in \mathcal{Y}$, we have:*

$$x = Q(y) \quad \text{if and only if} \quad y \in \partial h(x) . \tag{B.10}$$

(*b*) *The image* im $Q$ *of* $Q$ *satisfies* ri $\mathcal{C} \subseteq$ im $Q = $ dom $\partial h \subseteq \mathcal{C}$.

(*c*) *The convex conjugate* $h^*\colon \mathcal{Y} \to \mathbb{R}$ *of* $h$ *is differentiable and*

$$Q(y) = \nabla h^*(y) \quad \text{for all } y \in \mathcal{Y}. \tag{B.11}$$

(*d*) $Q$ *is* $(1/K)$-*Lipschitz continuous, that is,*

$$\|Q(y') - Q(y)\| \le (1/K)\|y' - y\|_* \quad \text{for all } y, y' \in \mathcal{Y}. \tag{B.12}$$

(*e*) *Fix some* $y \in \mathcal{Y}$ *and set* $x = Q(y)$. *Then, for all* $x' \in \mathcal{X}$ *we have:*

$$\partial h(x; x' - x) \ge \langle y, x' - x \rangle. \tag{B.13}$$

*In particular, if* $\partial h$ *admits a continuous selection* $\nabla h\colon$ dom $\partial h \to \mathcal{Y}$, *we have*

$$\langle \nabla h(x), x' - x \rangle \ge \langle y, x' - x \rangle \quad \text{for all } x \in \text{dom } \partial h \text{ and all } x \in \mathcal{C}, \tag{B.14}$$

*or, equivalently,*

$$\partial h(x) = \nabla h(x) + \text{PC}(x) \quad \text{for all } x \in \text{dom } \partial h, \tag{B.15}$$

*where*

$$\text{PC}(x) = \{w \in \mathcal{Y} : \langle w, x' - x \rangle \le 0 \ \text{ for all } x' \in \mathcal{X}\} \tag{B.16}$$

*denotes the* polar cone *to* $\mathcal{C}$ *at* $x$.

*Proof.* These properties are fairly well known (except possibly the last one), so we only provide a quick proof or a precise pointer to the literature.

(*a*) The maximum in (B.8) is attained for all $y \in \mathcal{V}^*$ and is unique because $h$ is strongly convex. Furthermore, $x$ solves (B.8) if and only if $y - \partial h(x) \ni 0$, i.e., if and only if $y \in \partial h(x)$.

(*b*) By (B.10), we readily get im $Q =$ dom $\partial h$. Consequently, the rest of our claim follows from standard results in convex analysis, see e.g., Rockafellar [52, Chap. 26].

(*c*) The equality $Q = \nabla h^*$ follows immediately from Danskin's theorem, see e.g., Bertsekas [4, Proposition 5.4.8, Appendix B].

(*d*) See Rockafellar & Wets [53, Theorem 12.60(b)].

(*e*) Since $y \in \partial h(x)$ by (B.10), we readily get that

$$h(x + \theta(x' - x)) \ge h(x) + \theta\langle y, x' - x \rangle \quad \text{for all } \theta \in [0, 1]. \tag{B.17}$$

Hence, by rearranging and taking the limit $\theta \to 0^+$,[10] we conclude that

$$\partial h(x; x' - x) = \lim_{\theta \to 0^+} \frac{h(x + \theta(x' - x)) - h(x)}{\theta} \ge \langle y, x' - x \rangle \tag{B.18}$$

as claimed. Finally, for our last assertion, let $z = x' - x$ and set

$$\phi(\theta) = h(x + \theta z) - [h(x) + \langle y, \theta z \rangle] \quad \text{for all } \theta \in [0, 1] \tag{B.19}$$

so $\phi(\theta) \ge K\theta^2\|z\|^2/2 \ge 0$ for all $\theta \in [0, 1]$. By construction, it is straightforward to verify that the function $\psi(\theta) = \langle \nabla h(x + \theta z) - y, z \rangle$ is a selection of subgradients of $\phi$, i.e.,

$$\phi(\theta') \ge \phi(\theta) + \psi(\theta)(\theta' - \theta) \quad \text{for all } \theta, \theta' \in [0, 1]. \tag{B.20}$$

Since $\psi$ is in addition continuous (because $\nabla h$ is), it follows that $\phi'(\theta) = \psi(\theta)$ for all $\theta \in [0, 1]$ by a well-known characterization of the one-sided derivatives of convex functions, cf. Rockafellar [52, Theorem 24.2]. Hence, with $\phi$ convex and $\phi(\theta) \ge \phi(0)$ for all $\theta \in [0, 1]$, we conclude that $\langle \nabla h(x) - y, z \rangle = \psi(0) = \phi'(0) \ge 0$, and our proof is complete. ∎

The next proposition collects some basic properties of the Fenchel coupling.

---

[10]The existence of the limit is guaranteed by standard results, see e.g., Bertsekas [4, Appendix B].

**Proposition B.2.** *Let $h$ be a $K$-strongly convex regularizer on $\mathcal{C}$. Then, for all $p \in \mathcal{X}$ and all $y, y' \in \mathcal{Y}$, we have:*

$(a)$ $F(p, y) \geq 0$ *with equality if and only if $p = Q(y)$.* $\qquad\qquad$ (B.21a)

$(b)$ $F(p, y) \geq \frac{1}{2}K \|Q(y) - p\|^2$. $\qquad\qquad$ (B.21b)

*Proof.* These properties are also fairly standard, but we provide a quick proof for completeness.

$(a)$ By the Fenchel–Young inequality, we have $h(p) + h^*(y) \geq \langle y, p \rangle$ for all $p \in \mathcal{X}, y \in \mathcal{Y}$, with equality if and only if $y \in \partial h(p)$. Our claim then follows from (B.10).

$(b)$ Let $x = Q(y)$ so $y \in \partial h(x)$ by (B.10). Then, by the definition of $F$, we have

$$\begin{aligned}
F(p, y) &= h(p) + h^*(y) - \langle y, p \rangle \\
&= h(p) + \langle y, x \rangle - h(x) - \langle y, p \rangle && \text{\% since } y \in \partial h(x) \\
&\geq h(p) - h(x) - \partial h(x; p - x) && \text{\% by Proposition B.1} \\
&\geq \tfrac{1}{2}K\|x - p\|^2 && \text{\% by (B.4)}
\end{aligned}$$

and our proof is complete. $\qquad\qquad\blacksquare$

In view of Proposition B.2, $F(p, y)$ can be seen a "primal-dual" measure of divergence between $p \in \mathcal{X}$ and $y \in \mathcal{Y}$, and the alternate expression (19) is straightforward. This observation will play a major role in the sequel.

**B.2. Basic lemmas.** Moving forward, we note that the various update steps in (FTRL+) can be written as

$$y^+ = y + w \quad \text{and} \quad x^+ = Q(y^+) \qquad\qquad (B.22)$$

for some $y, w \in \mathcal{Y}$. With this in mind, we proceed below to state a series of basic lemmas for the Fenchel coupling before and after an update of the form (B.22). These results are not new, cf. [31, 40, 42] and references therein; however, the assumptions used to derive them vary significantly in the literature, so we provide detailed proofs for completeness.

All of the results that follow below are stated for a $K$-strongly convex regularizer on $\mathcal{C}$. The first result is a primal-dual version of the so-called "three-point identity" for mirror descent [7]:

**Lemma B.1.** *Fix some $p \in \mathcal{X}$, $y \in \mathcal{Y}$, and let $x = Q(y)$. Then, for all $y^+ \in \mathcal{Y}$, we have:*

$$F(p, y^+) = F(p, y) + F(x, y^+) + \langle y^+ - y, x - p \rangle. \qquad\qquad (B.23)$$

*Proof.* By definition, we have:

$$\begin{aligned}
F(p, y^+) &= h(p) + h^*(y^+) - \langle y^+, p \rangle && (B.24a) \\
F(p, y) &= h(p) + h^*(y) - \langle y, p \rangle && (B.24b) \\
F(x, y^+) &= h(x) + h^*(y^+) - \langle y^+, x \rangle && (B.24c)
\end{aligned}$$

Thus, subtracting (B.24b) and (B.24c) from (B.24a), and rearranging, we get

$$F(p, y^+) = F(p, y) + F(x, y^+) - h(x) - h^*(y) + \langle y^+, x \rangle - \langle y^+ - y, p \rangle . \qquad (B.25)$$

Our assertion then follows by recalling that $x = Q(y)$, so $h(x) + h^*(y) = \langle y, x \rangle$. $\qquad\blacksquare$

The next result we present concerns the Fenchel coupling before and after a direct update step; similar results exist in the literature, but we again provide a proof for completeness.

**Lemma B.2.** *Fix some $p \in \mathcal{X}$ and $y, w \in \mathcal{Y}$. Then, letting $x = Q(y)$, $y^+ = y + w$, and $x^+ = Q(y^+)$ as per (B.22), we have:*

$$\begin{aligned}
F(p, y^+) &= F(p, y) + \langle w, x^+ - p \rangle - F(x^+, y) && (B.26a) \\
&\leq F(p, x) + \langle w, x - p \rangle + \tfrac{1}{2}K\|w\|_*^2 . && (B.26b)
\end{aligned}$$

*Proof.* By the three-point identity (B.23), we have

$$F(x, y) = F(x, y^+) + F(x^+, x) + \langle y - y^+, x^+ - p \rangle \tag{B.27}$$

so our first claim follows by rearranging. For our second claim, simply note that

$$F(p, y) + \langle w, x^+ - p \rangle - F(x^+, y) = F(p, y) + \langle w, x - p \rangle + \langle w, x^+ - x \rangle - F(p, y)$$
$$\leq F(p, y) + \langle w, x - p \rangle + \frac{1}{2K} \|w\|_*^2 + \frac{K}{2} \|x - p\|^2 - F(p, y) \tag{B.28}$$

so our claim follows from Proposition B.2. ∎

The last result we present here is sometimes referred to as a "four-point lemma", and concerns the Fenchel coupling before and after an *extrapolation* step:

**Lemma B.3.** *Fix some $p \in \mathcal{X}$ and $y, w_1, w_2 \in \mathcal{Y}$. Then, letting $x = Q(y)$, $y_i^+ = y + w_i$, and $x_i^+ = Q(y_i^+)$, $i = 1, 2$, as per (B.22), we have:*

$$F(p, y_2^+) = F(p, y) + \langle w_2, x_1^+ - p \rangle + \left[ \langle w_2, x_2^+ - x_1^+ \rangle - F(x_2^+, y) \right] \tag{B.29a}$$

$$= F(p, y) + \langle w_2, x_1^+ - p \rangle + \langle w_2 - w_1, x_2^+ - x_1^+ \rangle - F(x_2^+, y_1^+) - F(x_1^+, y) \tag{B.29b}$$

$$\leq F(p, y) + \langle w_2, x_1^+ - p \rangle + \frac{1}{2K} \|w_2 - w_1\|_*^2 - \frac{K}{2} \|x_1^+ - x\|^2 . \tag{B.29c}$$

*Proof.* By Lemma B.2, we have

$$F(p, y_2^+) = F(p, y) + \langle w_2, x_2^+ - p \rangle - F(x_2^+, y) \tag{B.30}$$

so (B.29a) follows by writing $\langle w_2, x_2^+ - p \rangle = \langle w_2, x_1^+ - p \rangle + \langle w_2, x_2^+ - x_1^+ \rangle$, and (B.29b) follows from the three-point identity (B.23) for the Fenchel coupling. Finally, for (B.29c), the Fenchel-Young inequality in Peter-Paul form yields

$$\langle w_2 - w_1, x_2^+ - x_1^+ \rangle \leq \frac{1}{2K} \|w_2 - w_1\|_*^2 + \frac{K}{2} \|x_2^+ - x_1^+\|^2 \tag{B.31}$$

and our claim follows again by invoking Proposition B.2 to write

$$\frac{K}{2} \|x_2^+ - x_1^+\|^2 - F(x_2^+, y_1^+) - F(x_1^+, y) \leq -F(x_1^+, y) \leq -\frac{K}{2} \|x_1^+ - x\|^2 \tag{B.32}$$

and then substituting the result in (B.31) ∎

Lemmas B.2 and B.3 will be responsible for most of the heavy lifting to derive a Lyapunov function for (FTRL+). We discuss the relevant details in Appendix D.

We conclude this section with a variational characterization of the abstract update (B.22) in the case where $\partial h$ of $h$ admits a continuous selection – or, alternatively, $h$ is smooth in the sense of (17).

**Lemma B.4.** *Fix some $y, y^+ \in \mathcal{Y}$, and let $x^+ = Q(y^+)$. Then, for all $p \in \mathcal{X}$, we have*

$$\langle y^+ - y, p - x^+ \rangle \leq \langle \nabla h(x^+) - y, p - x^+ \rangle . \tag{B.33}$$

*Proof.* Invoking (B.14) in Proposition B.1 with $y \leftarrow y^+$, $x \leftarrow x^+$, and $x' \leftarrow p$, we get

$$\langle y^+, p - x^+ \rangle \leq \langle \nabla h(x^+), p - x^+ \rangle . \tag{B.34}$$

Our claim then follows by subtracting $\langle y, p - x^+ \rangle$ from both sides of the above. ∎

## C  Continuous-time analysis

**C.1. Dynamical systems notions.** To fix notation, we recall here some basics from the theory of dynamical systems, roughly following [2, 51]. In this section, $\mathcal{M}$ is an open subset of a Euclidean space of dimension $d$.

We consider a system of ordinary differential equations (ODEs) of the form

$$\dot{x}(t) = X(x(t)), \tag{DS}$$

where $x(t)$ is a curve in $\mathcal{M}$ defined on an open interval $\mathcal{I} \subseteq \mathbb{R}$ (that without loss of generality we assume to contain 0), and $X \colon \mathcal{M} \to \mathbb{R}^d$ is a smooth function. The function $X$ is called *vector field* because it assigns a vector $X(x)$ to each point $x$ in $\mathcal{M}$, and (DS) is called *dynamical system generated by $X$*.

Given $x_0 \in \mathcal{M}$, an *orbit with initial condition $x_0$* is a solution $x(t)$ of (DS) with $x(0) = x_0$. The *flow generated by $X$* is the smooth function $\Theta \colon \mathcal{I} \times \mathcal{M} \to \mathcal{M}$ such that $\Theta_0(x_0) = x_0$ for all $x_0 \in \mathcal{M}$ and $\frac{d}{dt}\Theta_t(x) = X(\Theta_t(x))$ for all $t \in \mathcal{I}$. In words, $\Theta_t(x_0)$ is the orbit $x(t)$ with initial condition $x_0$; the existence and uniqueness of this function is guaranteed by the existence and uniqueness theorem of solutions of ordinary differential equations.

A flow $\Theta$ is called *volume-preserving* if $\mathrm{vol}(\Theta_t(\mathcal{U})) = \mathrm{vol}(\mathcal{U})$ for any (Lebesgue) measurable subset $\mathcal{U} \subseteq \mathcal{M}$ and all $t \in \mathcal{I}$. Liouville's theorem gives a sufficient condition for a flow to be volume-preserving based on the *divergence* of its generating field:[11]

**Theorem** (Liouville). *If $\mathrm{div}\, X \equiv 0$ then the flow generated by $X$ is volume-preserving.*

Volume-preserving flows are closely related to recurrent dynamical patterns. A point $x \in \mathcal{M}$ is said to be *recurrent* under (DS) if, for every neighborhood $\mathcal{U}$ of $x \in \mathcal{M}$, there exists an increasing sequence of time $t_n \uparrow \infty$ such that $\Theta_{t_n}(x)$ is defined and falls in $\mathcal{U}$ for all $n$. Moreover, (DS) is said to be *Poincaré recurrent* if almost every point $x \in \mathcal{M}$ is recurrent. The celebrated Poincaré recurrence theorem gives a sufficient condition for a dynamical system to be Poincaré recurrent:

**Theorem** (Poincaré). *Let $X$ be a smooth vector field on $\mathcal{M}$. If the flow induced by $X$ is volume-preserving and all the orbits of (DS) are bounded, then (DS) is Poincaré recurrent.*

**C.2. Basic properties of FTRL.** In this section we survey some of the properties of the follow-the-regularized-leader learning scheme in a continuous-time, multi-agent setting, in line with the presentations of [16, 38, 41]. For ease of reference we recall here some of the notions introduced in Appendix B and in Sections 2 and 3 from the main body of the paper.

Let $\Gamma = \Gamma(\mathcal{N}, \mathcal{A}, u)$ be a finite normal form game, and let $v$ denote its payoff field. The game's strategy space is $\mathcal{X} = \prod_{j \in \mathcal{N}} \Delta(\mathcal{A}_j) \subseteq \mathcal{V} := \prod_j \mathbb{R}^{\mathcal{A}_j}$, and the game's payoff space is $\mathcal{Y} := \mathcal{V}^*$. The payoff field is a map $v \colon \mathcal{V} \to \mathcal{Y}$ that evaluated at a strategy $x \in \mathcal{X}$ acts linearly on any $x' \in \mathcal{X}$ by

$$
\begin{aligned}
\langle v(x), x' \rangle &= \sum_{i \in \mathcal{N}} \langle v_i(x), x_i' \rangle = \sum_{i \in \mathcal{N}} \sum_{\alpha_i \in \mathcal{A}_i} v_{i\alpha_i}(x) x_{i\alpha_i}' \\
&= \sum_{i \in \mathcal{N}} u_i(x_i', x_{-i}) \in \mathbb{R}.
\end{aligned} \tag{C.1}
$$

Assume now that $\Gamma$ is played continuously over time. As discussed in Section 3, the main idea behind the follow-the-regularized-leader learning scheme is that, at any given time $t \geq 0$, each player $i \in \mathcal{N}$ tracks their cumulative payoff up to time $t$ and plays a "regularized" best response strategy in light of this information. Concretely, given a cumulative payoff vector $y_i(t) \in \mathcal{Y}_i$, each player $i \in \mathcal{N}$ selects this optimal strategy $x_i(t) \in \mathcal{X}_i$ by means of a *regularized best response* map $Q_i \colon \mathcal{Y}_i \to \mathcal{X}_i$, a single-valued analogue of the best-response correspondence $y_i \mapsto \arg\max_{x_i \in \mathcal{X}_i} \langle y_i, x_i \rangle$. A standard way [57] of obtaining such map is to introduce a *regularizer function* $h_i \colon \mathcal{X}_i \to \mathbb{R}$ that is *(i)* continuous on $\mathcal{X}_i$, *(ii)* smooth on $\mathrm{ri}\,\mathcal{X}_i$, the relative interior of $\mathcal{X}_i$, and *(iii)* strongly convex on $\mathcal{X}_i$ (as per Eq. (B.4)); and to consider the induced *choice map* $Q_i \colon \mathcal{Y}_i \to \mathcal{X}_i$ defined by

$$Q_i(y_i) = \arg\max_{x_i \in \mathcal{X}_i} \{ \langle y_i, x_i \rangle - h_i(x_i) \} \quad \text{for all } y_i \in \mathcal{Y}_i. \tag{6}$$

By Proposition B.1, $Q_i$ is well-defined and Lipschitz continuous, and it coincides with the differential $\nabla h_i^*$ of $h_i^* \colon \mathcal{Y}_i \to \mathbb{R}$, the *convex conjugate* of $h_i$.

---

[11]Recall here that the divergence is a differential operator mapping a vector field $X$ on $\mathcal{M}$ to the real-valued function $\mathrm{div}\, X(x) := \sum_{\alpha=1}^{d} \partial_\alpha X^\alpha(x)$, where $\partial_\alpha$ is a shorthand for the partial derivative $\partial/\partial x_\alpha$

In a continuous time setting, this regularized learning scheme translates into the following implicit equations of motion, which govern the evolution of the cumulative payoff $y(t) \in \mathcal{Y}$ and of the mixed strategy profile $x(t) \in \mathcal{X}$ as the players attempt to maximize their payoff over time:

$$y_{i\alpha_i}(t) = y_{i\alpha_i}(0) + \int_0^t v_{i\alpha_i}(x(\tau)) \, d\tau \quad \text{with} \quad x_i(t) = Q_i(y_i(t)), \tag{C.2}$$

for all $i \in \mathcal{N}, \alpha_i \in \mathcal{A}_i$. A straightforward computation shows that this is equivalent to Eq. (5) from Section 3 in the main text, that governs the evolution of the mixed strategy $x(t) \in \mathcal{X}$:

$$x_i(t) = \arg\max_{p_i \in \mathcal{X}_i} \left\{ \int_0^t u_i(p_i; x_{-i}(\tau)) \, d\tau - h_i(p_i) \right\} = \arg\max_{p_i \in \mathcal{X}_i} \left\{ \int_0^t \langle v_i(x(\tau)), p_i \rangle \, d\tau - h_i(p_i) \right\}. \tag{5}$$

Importantly, Eq. (C.2) can be cast in the form (DS) of a dynamical system in the game's payoff space. For each $i \in \mathcal{N}$, differentiation with respect to $t$ yields

$$\dot{y}_i(t) = v_i(x(t)) \qquad x_i(t) = Q_i(y_i(t)), \tag{FTRL-D}$$

and by aggregating the player indices we obtain the system of ODEs

$$\dot{y} = Y(y), \tag{C.3}$$

where $Y := v \circ Q : \mathcal{Y} \to \mathcal{Y}$ is a continuous vector field on $\mathcal{Y}$; cf. Fig. 3.

Existence and uniqueness of a global solution $y(t) \in \mathcal{Y}$ of Eq. (C.3) for any initial condition $y(0) \in \mathcal{Y}$ are guaranteed by standard arguments [38, Prop. 3.1]; in line with the terminology of the previous section we will refer to such a solution as a *dual orbit*.

**C.3. Constant of motion for harmonic games.** The following result shows that FTRL in harmonic games admits a constant of motion.

**Proposition C.1.** *Let $\Gamma = \Gamma(\mathcal{N}, \mathcal{A}, u)$ be a finite game and consider a vector $m \in \mathbb{R}_{++}^N$ and a fully mixed strategy $q \in \mathcal{X}$. Then the weighted Fenchel coupling $F_{m,q} : \mathcal{Y} \to \mathbb{R}$ defined by*

$$F_{m,q}(y) := \sum_i m_i F_i(q_i, y_i) = \sum_i m_i \left( h_i(q_i) + h_i^*(y_i) - \langle q_i, y_i \rangle \right) \tag{C.4}$$

*is a constant of motion under (FTRL-D) if and only if $\Gamma$ is harmonic with strategic center $(m, q)$.*

*Proof.* Let $y(t)$ be a dual orbit. Then by chain rule

$$\frac{d}{dt} F_{m,q}(y(t)) = \sum_i m_i \left[ \langle \nabla h_i^*(y_i), \dot{y}_i \rangle - \langle q_i, \dot{y}_i \rangle \right] = \sum_i m_i \langle x_i(t) - q_i, v_i(x(t)) \rangle \tag{C.5}$$

where the second equality holds by (FTRL-D) and Eq. (B.11). Then, by the characterization of harmonic games in terms of a strategic center (HG-center), the time derivative of the weighted Fenchel coupling vanishes identically along a dual orbit of (FTRL-D) precisely if the underlying game is harmonic. ∎

The existence of this constant of motion is fundamental for proving Theorem 2, i.e., the Poincaré recurrence of continuous-time FTRL in harmonic games. With this key element established, the remainder of this appendix closely follows the proof technique described by [41] for the analogous result in the context of two-player zero-sum games.

**C.4. FTRL in the space of payoff differences.** For any initial condition $y(0) \in \mathcal{Y}$, a dual orbit of (FTRL-D) induces a curve $x(t) = Q(y(t))$ in the game's strategy space $\mathcal{X}$ which solves Eq. (5) for all $t \geq 0$; in the following we will refer to such curve as *trajectory of play*. Crucially, a trajectory of play is in general *not* the global solution of a dynamical system $\dot{x} = X(x)$ for some vector field $X : \mathcal{X} \to \mathcal{X}$ in the game's strategy space. The reason for this is that the map $Q : \mathcal{Y} \to \mathcal{X}$ is not necessarily invertible, so there is in general no way to identify a unique a vector field $X$ on $\mathcal{X}$ that is *related* to the vector field $Y$ on $\mathcal{Y}$ via $Q$.

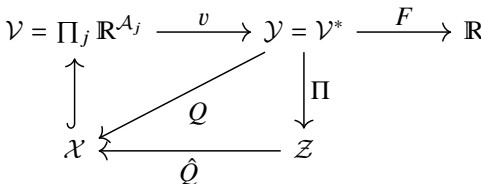

**Figure 3: FTRL diagram.** Commutative diagram of the maps discussed in Appendices C.2–C.4; note in particular that $v \circ Q$ is a vector field on $\mathcal{Y}$. The notation $\mathcal{X} \hookrightarrow \mathcal{V}$ is equivalent to $\mathcal{X} \subseteq \mathcal{V}$.

**Related vector fields and induced dynamical systems.** The concept of vector fields related by a smooth map is standard in differential geometry (e.g., [34, p. 181]). Let $\mathcal{M}, \mathcal{M}'$ be open subsets of Euclidean space: given a vector field $Y$ on $\mathcal{M}$ and a smooth map $F \colon \mathcal{M} \to \mathcal{M}'$, a vector field $X$ on $\mathcal{M}'$ is called $F$-*related to* $Y$ if, for all $y \in \mathcal{M}$, $(\mathrm{Jac}\, F)_y \cdot Y(y) = X(x)$, with $x = F(y)$. Here $\mathrm{Jac}\, F$ is the Jacobian matrix of $F$, and $\cdot$ represents matrix-vector multiplication. If $F$ is invertible then such vector field exists always and is unique; else, it might exist and not be unique, or not exist at all.

Vector fields that are related via a smooth map are useful inasmuch as they generate "compatible" dynamical systems:

**Lemma C.2.** *Let $F \colon \mathcal{M} \to \mathcal{M}'$ be a smooth map between open subsets of Euclidean spaces, and let $\dot{y} = Y(y)$ be a dynamical system on $\mathcal{M}$. Let $y(t)$ be an orbit with initial condition $y_0 \in \mathcal{M}$, and consider the curve on $\mathcal{M}'$ defined by $x(t) := F(y(t))$. If there exists a vector field $X$ on $\mathcal{M}'$ that is $F$-related to $Y$, then the curve $x(t)$ is an orbit of the dynamical system $\dot{x} = X(x)$ with initial condition $x_0 = F(y_0)$.*

*Proof.* By chain rule,

$$\frac{d}{dt}x(t) = \frac{d}{dt}F(y(t)) = (\mathrm{Jac}\, F)_{y(t)} \cdot \dot{y}(t) = (\mathrm{Jac}\, F)_{y(t)} \cdot Y(y(t)) = X(x(t)), \qquad (\mathrm{C.6})$$

where the last equality follows by the assumption that $X$ is $F$-related to $Y$. $\blacksquare$

In the following, if $F \colon \mathcal{M} \to \mathcal{M}'$ is a smooth function between open subsets of Euclidean spaces, and $Y, X$ are vector fields fulfilling the assumptions of Lemma C.2, we say that the dynamical system $\dot{y} = Y(y)$ on $\mathcal{M}$ *induces the dynamical system* $\dot{x} = X(x)$ *on* $\mathcal{M}'$ *via* $F$.

**FTRL induced in the space of payoff differences.** The choice map $Q \colon \mathcal{Y} \to \mathcal{X}$ is in general not smooth, and neither injective nor surjective [16, Sec.3], so it generally does not allow to induce the dynamical system (C.3) from the game's payoff space $\mathcal{Y}$ to the game's strategy space $\mathcal{X}$. [12] In other words, the learning process (FTRL-D) in a finite game gives rise to a dynamical system in the game's payoff space $\mathcal{Y}$, to which the theorems presented in Appendix C.1 can in principle be applied; however, it can be shown that the orbits of Eq. (C.3) in $\mathcal{Y}$ are *not* bounded, preventing the application of Poincaré's theorem. Furthermore, the dual orbits do not convey direct information on the day-to-day behavior of the players, due to the lack of invertibility of the choice map.

Conversely, the objects of interest from a dynamical, learning viewpoint – that is, the trajectories of play in the game's strategy space $\mathcal{X}$ – present technical difficulties and do not easily fit the dynamical systems framework depicted in Appendix C.1. In the following we show how these difficulties can be circumvented by analyzing the dynamics induced by (FTRL-D) in yet a third space $\mathcal{Z}$, that arises by taking the *differences* between payoffs – rather than their absolute values – as the objects of study.

To make this precise, given the game $\Gamma = \Gamma(\mathcal{N}, \mathcal{A}, u)$ fix a benchmark strategy $\hat{\alpha}_i \in \mathcal{A}_i$ for every player $i \in \mathcal{N}$, and consider the hyperplane $\mathcal{Z}_i := \{z_i \in \mathbb{R}^{A_i} : z_{i\hat{\alpha}_i} = 0\} \subset \mathbb{R}^{A_i}$. Clearly, $\mathcal{Z}_i \cong \mathbb{R}^{A_i-1}$. Each player's strategy space $\mathcal{Y}_i = \mathbb{R}^{A_i}$ can be mapped onto $\mathcal{Z}_i$ by the linear operator

$$\Pi_i \colon \mathcal{Y}_i \to \mathcal{Z}_i \quad \text{with} \quad z_{i\alpha_i} = y_{i\alpha_i} - y_{i\hat{\alpha}_i} \qquad (\mathrm{C.7})$$

for all $\alpha_i \in \mathcal{A}_i$.

---

[12] A detailed treatment of the conditions under which a trajectory of play $x(t)$ actually *is* a solution of dynamical system in the game's strategy space $\mathcal{X}$ is beyond the scope of this work; we refer the interested reader to [38, 39] for an in-depth treatment.

$\Pi_i$ is clearly smooth, and a standard check shows that $\Pi_i$ is surjective and not injective: $\ker \Pi_i = \{y_i : y_{i\alpha_i} = y_{i\beta_i}$ for all $\alpha_i, \beta_i \in \mathcal{A}_i\}$ is the 1-dimensional linear subspace spanned by the vector $\mathbf{1}_i = (1, \dots, 1) \in \mathcal{Y}_i$; and $\Pi^{-1}(z_i) = z_i + \ker \Pi_i$ for any $z_i \in \mathcal{Z}_i$. In particular, for all $y_i, y_i' \in \mathcal{Y}_i$, we have that $\Pi_i(y_i) = \Pi_i(y_i')$ if and only if $y_i - y_i'$ is proportional to $\mathbf{1}_i$.

Since every $z_i \in \mathcal{Z}_i$ is the image of some payoff $y_i$ via $\Pi_i$, the space $\mathcal{Z} := \prod_j \mathcal{Z}_j$ is called the game's *payoff-difference space*; we will denote by $\Pi$ the product map $\Pi \equiv \prod_j \Pi_j$, i.e., (cf. Fig. 3)

$$\Pi : \mathcal{Y} \to \mathcal{Z}, \quad \Pi(y) := (\Pi_i(y_i))_{i \in \mathcal{N}}. \tag{C.8}$$

**Lemma C.3.** *The choice map $Q : \mathcal{Y} \to \mathcal{X}$ is invariant on the level sets of $\Pi$.*

*Proof.* Let $y, y' \in \mathcal{Y}$. By the discussion above, $\Pi(y) = \Pi(y')$ iff $y_i' - y_i = \lambda \mathbf{1}_i$ for some $\lambda \in \mathbb{R}$. Then for each $i \in \mathcal{N}$,

$$Q_i(y_i') = \underset{x_i \in \mathcal{X}_i}{\arg\max}\{\langle y_i', x_i \rangle - h_i(x_i)\} = \underset{x_i \in \mathcal{X}_i}{\arg\max}\{\langle y_i, x_i \rangle + \lambda \langle \mathbf{1}_i, x_i \rangle - h_i(x_i)\} = Q_i(y_i). \quad \blacksquare$$

**Proposition C.4.** *The dynamical system (C.3) in the game's payoff space $\mathcal{Y}$ induces a dynamical system*

$$\dot{z} = Z(z) \tag{C.9}$$

*in the game's payoff-difference space $\mathcal{Z}$ via the map (C.8).*

*Proof.* By the discussion in the previous section (and in particular Lemma C.2), if we exhibit a vector field $Z$ on $\mathcal{Z}$ that is $\Pi$-related to $Y$, then our proof is complete. Thus we look for a vector field $Z$ such that, for all $y \in \mathcal{Y}$,

$$(\operatorname{Jac} \Pi)_y \cdot Y(y) = Z(z), \tag{C.10}$$

with $z = \Pi(y)$. By Eq. (C.7), $(\operatorname{Jac} \Pi_i)_{\alpha_i \beta_i} = \delta_{\alpha_i \beta_i} - \delta_{\hat{\alpha}_i \beta_i}$. Since $Y = v \circ Q$, the sought-after vector field $Z$ must fulfill, for all $y \in \mathcal{Y}$ and all $\alpha_i \in \mathcal{A}_i$,

$$(v_{i\alpha_i} - v_{i\hat{\alpha}_i}) \circ Q_i(y_i) = Z_{i\alpha_i}(z_i), \tag{C.11}$$

with $z = \Pi(y)$. For each $i \in \mathcal{N}$ define now (cf. Fig. 3)

$$\hat{Q}_i : \mathcal{Z}_i \to \mathcal{X}_i, \quad \hat{Q}_i(z_i) = Q(y_i) \tag{C.12}$$

for *any* $y_i \in \Pi_i^{-1}(z_i)$, and denote by $\hat{Q} : \mathcal{Z} \to \mathcal{X}$ the induced product map. Such map exists since $\Pi_i$ is surjective, and is well-defined by Lemma C.3. It follows that the vector field on $\mathcal{Z}$ defined by

$$Z_{i\alpha_i}(z_i) := (v_{i\alpha_i} - v_{i\hat{\alpha}_i}) \circ \hat{Q}_i(z_i) \tag{C.13}$$

for all $i \in \mathcal{N}, z_i \in \mathcal{Z}_i, \alpha_i \in \mathcal{A}_i$ fulfills Eq. (C.11), and is hence $\Pi$-related to $Y$. $\quad \blacksquare$

This result shows that, for every dual orbit $y(t)$ of Eq. (C.3) with initial condition $y_0 \in \mathcal{Y}$, the curve $z(t) = \Pi(y(t))$ is an orbit of the dynamical system (C.9) in $\mathcal{Z}$ with initial condition $\Pi(y_0)$. To conclude this section we give a result implying that if the weighted Fenchel coupling (C.4) is a constant of motion constant then the solution orbits of (C.9) in $\mathcal{Z}$ are bounded.

**Lemma C.5.** *For any $i \in \mathcal{N}$, let $y_{i,n}$ be a sequence in $\mathcal{Y}_i$, and let $p_i$ be a point in the relative interior of $\mathcal{X}_i$. If $\sup_n |h_i^*(y_{i,n}) - \langle y_{i,n}, p_i \rangle| < \infty$, then also the score differences remain bounded, i.e., $|y_{i\alpha_i,n} - y_{i\beta_i,n}| < \infty$ for all $\alpha_i, \beta_i \in \mathcal{A}_i$ and all $n$.*

*Proof.* See [41, Appendix D]. $\quad \blacksquare$

**Lemma C.6.** *If the weighted Fenchel coupling (C.4) is a constant of motion under (FTRL-D) for some fully mixed $p \in \mathcal{X}$ then the orbits of $\dot{z} = Z(z)$ as in Eq. (C.9) are bounded in $\mathcal{Z}$.*

*Proof.* Assume that $F_{m,q}(y) = \sum_i m_i F_i(p_i, y_i) = \sum_i m_i \left(h_i(p_i) + h_i^*(y_i) - \langle p_i, y_i \rangle\right)$ is a constant of motion for (FTRL-D) for some fully mixed $p \in \mathcal{X}$ and some $m \in \mathbb{R}_{++}^N$. Let $y(t)$ be an orbit of (FTRL-D) in $\mathcal{Y}$, and let $y_{i,n} := y_i(t_n)$ for any sequence of time $t_n$. Let furthermore $F_{i,n} := h_i^*(y_{i,n}) - \langle p_i, y_{i,n} \rangle$. Then $\sup_n |F_{i,n}| < \infty$. By Lemma C.5, this implies that $|z_{i\alpha_i,n}| < \infty$ for all $\alpha_i \in \mathcal{A}_i$, all $i \in \mathcal{N}$, and all $n$. $\quad \blacksquare$

**C.5. Recurrence of FTRL in harmonic games.** We now have all the ingredients to prove that almost every trajectory of play $x(t)$ of (FTRL-D) in harmonic games returns arbitrarily close to its starting point infinitely often.

**Theorem 2.** *Suppose $\Gamma$ is harmonic. Then almost every orbit $x(t)$ of* (FTRL-D) *returns arbitrarily close to its starting point infinitely often: specifically, for* (*Lebesgue*) *almost every initial condition $x(0) = Q(y(0)) \in \mathcal{X}$, there exists an increasing sequence of times $t_n \uparrow \infty$ such that $x(t_n) \to x(0)$.*

*Proof of Theorem 2.* The proof relies on the following steps:

1. the vector field $Z$ defined in Eq. (C.13) has vanishing divergence, so its induced flow is volume-preserving in $\mathcal{Z}$ by Liouville's theorem;

2. the orbits of the dynamical system $\dot{z} = Z(z)$ of Eq. (C.9) are bounded in $\mathcal{Z}$ since the weighted Fenchel coupling (C.4) is a constant of motion for FTRL in harmonic games;

3. the dynamical system $\dot{z} = Z(z)$ is recurrent in $\mathcal{Z}$ by Poincaré theorem;

4. by continuity of Eq. (C.12), almost every trajectory of play $x(t)$ of (FTRL-D) with initial condition in the image of $\hat{Q}$ returns arbitrarily close to its starting point infinitely often.

Indeed, $\operatorname{div} Z(z) = \sum_i \sum_{\alpha_i} \frac{\partial}{\partial z_{i\alpha_i}} ((v_{i\alpha_i} - v_{i\hat{\alpha}_i}) \circ \hat{Q}_i(z_i))$. For the first term, by chain rule,

$$\operatorname{div} Z(z) = \sum_i \sum_{\alpha_i} \frac{\partial v_{i\alpha_i}}{\partial z_{i\alpha_i}}(\hat{Q}_i(z_i)) = \sum_i \sum_{\alpha_i} \sum_j \sum_{\beta_j} \frac{\partial v_{i\alpha_i}}{\partial x_{j\beta_j}}(\hat{Q}(z)) \frac{\partial \hat{Q}_{j\beta_j}}{\partial z_{i\alpha_i}}(z)$$

$$= \sum_i \sum_{\alpha_i} \sum_{\beta_i} \frac{\partial v_{i\alpha_i}}{\partial x_{i\beta_i}}(\hat{Q}(z)) \frac{\partial \hat{Q}_{i\beta_i}}{\partial z_{i\alpha_i}}(z) \equiv 0$$

since $\frac{\partial v_{i\alpha_i}}{\partial x_{i\beta_i}} \equiv 0$ by multilinearity of the payoff functions. The second term yields identical result with $\hat{\alpha}_i \leftarrow \alpha_i$, so we conclude that $\operatorname{div} Z = 0$. By Lemma C.6, the invariance of the weighted Fenchel coupling under (FTRL-D) implies that the payoff differences $z_{i\alpha_i}(t) = y_{i\alpha_i}(t) - z_{i\hat{\alpha}_i}(t)$ remain bounded for all $t \in [0, \infty)$. So, by Poincaré theorem, the dynamical system $\dot{z} = Z(z)$ is Poincaré recurrent, i.e., there exists a sequence of time $t_n \uparrow \infty$ such that $\lim_{n\to\infty} z(t_n) = z_0$ for almost every $z_0 \in \mathcal{Z}$. By continuity of (C.12), almost every trajectory of play $x(t) = Q(y(t)) = \hat{Q}(z(t))$ with $x_0 \in \operatorname{im} \hat{Q}$ fulfills $\lim_{n\to\infty} x(t_n) = x_0$, which concludes our proof by noting that the image of $\hat{Q}$ is the same as the image of $Q$. ∎

# D   Discrete-time analysis

In this appendix, our aim is to provide the full proofs for the discrete-time guarantees of (FTRL+), as presented in Section 4. Our analysis hinges on a series of energy functions and associated template inequalities, which we introduce in the next section.

**D.1. Lyapunov functions and template inequalities for** (FTRL+). The main building block of our analysis is a suitable Lyapunov function for the discrete-time algorithmic template (FTRL+). Motivated by the continuous-time analysis of Appendix C, we begin by considering the player-specific Fenchel couplings

$$F_i(p_i, y_i) = h_i(p_i) + h_i^*(y_i) - \langle y_i, p_i \rangle \quad \text{for } p_i \in \mathcal{X}_i, y_i \in \mathcal{Y}_i \tag{D.1}$$

induced by the regularizer $h_i$ of player $i \in \mathcal{N}$.

Suppose now that the game is harmonic relative to some measure $\mu = (\mu_1, \ldots, \mu_N)$, let $m_i = \sum_{\alpha_i \in \mathcal{A}_i} \mu_{i\alpha_i}$ denote the mass of $\mu_i$, and assume further that each player is running (FTRL+) with learning rate $\eta_i > 0$. Our analysis will hinge on the energy function

$$E(p, y) = \sum_{i \in \mathcal{N}} \frac{m_i}{\eta_i} F_i(p_i, y_i) \qquad p \in \mathcal{X}, y \in \mathcal{Y}, \tag{18}$$

which, as we show below, satisfies the following template inequality:

**Proposition D.1.** *Suppose that each player is running* (FTRL+) *with learning rate* $\eta_i > 0$ *in a harmonic game as above. Then, for all* $p_i \in \mathcal{X}_i$, $i \in \mathcal{N}$, *and all* $n = 1, 2, \ldots,$ *the algorithm's energy* $E_n := E(p, y_n)$ *enjoys the iterative bound:*

$$
\begin{aligned}
E_{n+1} \le E_n &+ \sum_{i \in \mathcal{N}} m_i \langle v_i(x_{n+1/2}), x_{i,n+1/2} - p_i \rangle \\
&+ \sum_{i \in \mathcal{N}} m_i \langle v_i(x_{n+1/2}) - v_i(x_n), x_{i,n+1} - x_{i,n+1/2} \rangle \\
&+ \sum_{i \in \mathcal{N}} m_i (1 - \lambda_i) \langle v_i(x_n) - v_i(x_{n-1/2}), x_{i,n+1} - x_{i,n+1/2} \rangle \\
&- \sum_{i \in \mathcal{N}} \frac{m_i}{\eta_i} F_i(x_{i,n+1}, y_{i,n+1/2}) \\
&- \sum_{i \in \mathcal{N}} \frac{m_i}{\eta_i} F_i(x_{i,n+1/2}, y_{i,n}) \,.
\end{aligned}
\tag{D.2}
$$

*Proof.* We begin by applying the bound (B.29b) of Lemma B.3 with the array of substitutions

1. $p \leftarrow p_i$
2. $w_1 \leftarrow \eta_i \hat{v}_{i,n} = \eta_i \lambda_i \, v_i(x_n) + \eta_i (1 - \lambda_i) \, v_i(x_{n-1/2})$
3. $w_2 \leftarrow \eta_i \hat{v}_{i,n+1/2} = \eta_i v_i(x_{n+1/2})$
4. $y \leftarrow y_{i,n}$      so      $x \leftarrow Q_i(y_{i,n}) = x_{i,n}$
5. $y_1^+ \leftarrow y_{i,n+1/2}$ so      $x_1^+ \leftarrow x_{i,n+1/2}$
6. $y_2^+ \leftarrow y_{i,n+1}$    so      $x_2^+ \leftarrow x_{i,n+1}$

We then get

$$
\begin{aligned}
\langle w_2 - w_1, x_2^+ - x_1^+ \rangle &= \eta_i \langle v_i(x_{n+1/2}) - \lambda_i \, v_i(x_n) - (1 - \lambda_i) \, v_i(x_{n-1/2}), x_{i,n+1} - x_{i,n+1/2} \rangle \\
&= \eta_i \langle v_i(x_{n+1/2}) - v_i(x_n), x_{i,n+1} - x_{i,n+1/2} \rangle \\
&\quad + \eta_i (1 - \lambda_i) \langle v_i(x_n) - v_i(x_{n-1/2}), x_{i,n+1} - x_{i,n+1/2} \rangle
\end{aligned}
\tag{D.3}
$$

and hence, by (B.29b):

$$
\begin{aligned}
F_i(p_i, y_{i,n+1}) \le\ &F_i(p_i, y_{i,n}) + \eta_i \langle v_i(x_{n+1/2}), x_{i,n+1/2} - p_i \rangle \\
&+ \eta_i \langle v_i(x_{n+1/2}) - v_i(x_n), x_{i,n+1} - x_{i,n+1/2} \rangle \\
&+ \eta_i (1 - \lambda_i) \langle v_i(x_n) - v_i(x_{n-1/2}), x_{i,n+1} - x_{i,n+1/2} \rangle \\
&- F_i(x_{i,n+1}, y_{i,n+1/2}) \\
&- F_i(x_{i,n+1/2}, y_{i,n}) \,.
\end{aligned}
\tag{D.4}
$$

Accordingly, with $E_n = E(p, y_n)$, the bound (D.2) follows by multiplying both sides by $m_i/\eta_i$ and summing over $i \in \mathcal{N}$. ∎

Thanks to Proposition D.1, we are now in a position to state and prove the following summability guarantee for (FTRL+).

**Proposition D.2.** *Suppose that each player in a harmonic game* $\Gamma$ *with harmonic measure* $\mu$ *is following* (FTRL+) *with learning rate* $\eta_i \le m_i K_i [2(N + 2) \max_j m_j G_j]^{-1}$. *Then, for all* $T$, *we have:*

$$
\sum_{n=1}^{T} \|x_{n+1/2} - x_n\|^2 + \sum_{n=2}^{T} \|x_n - x_{n-1/2}\|^2 \le \frac{2E_1}{(N + 2) \max_i m_i G_i} \,.
\tag{D.5}
$$

*In particular, the sequences* $A_n := \|x_{n+1/2} - x_n\|^2$ *and* $B_n := \|x_{n+1} - x_{n+1/2}\|^2$ *are both summable.*

*Proof.* By reshuffling the terms of the template inequality (D.2), we get

$$
\sum_{i \in \mathcal{N}} m_i \langle v_i(x_{n+1/2}), p_i - x_{i,n+1/2} \rangle \le E_n - E_{n+1}
$$

$$
+ \sum_{i \in \mathcal{N}} m_i \langle v_i(x_{n+1/2}) - v_i(x_n), x_{i,n+1} - x_{i,n+1/2} \rangle
\tag{D.6a}
$$

$$+ \sum_{i \in \mathcal{N}} m_i(1 - \lambda_i)\langle v_i(x_n) - v_i(x_{n-1/2}), x_{i,n+1} - x_{i,n+1/2}\rangle \quad \text{(D.6b)}$$

$$- \sum_{i \in \mathcal{N}} \frac{m_i}{\eta_i} F_i(x_{i,n+1}, y_{i,n+1/2}) - \sum_{i \in \mathcal{N}} \frac{m_i}{\eta_i} F_i(x_{i,n+1/2}, y_{i,n}) \, . \quad \text{(D.6c)}$$

We now proceed to bound each term of (D.6) individually, paying no heed to make the resulting bounds as tight as possible.

**Bounding (D.6a).** By the Fenchel-Young inequality, we have:

$$\text{(D.6a)} \le \sum_{i \in \mathcal{N}} \frac{m_i}{2G_i} \|v_i(x_{n+1/2}) - v_i(x_n)\|_*^2 + \sum_{i \in \mathcal{N}} \frac{m_i G_i}{2} \|x_{i,n+1} - x_{i,n+1/2}\|^2$$

$$\le \sum_{i \in \mathcal{N}} \frac{m_i G_i}{2} \|x_{n+1/2} - x_n\|^2 + \sum_{i \in \mathcal{N}} \frac{m_i G_i}{2} \|x_{i,n+1} - x_{i,n+1/2}\|^2 \qquad \text{\% } v_i(x) \text{ is } G_i\text{-Lipschitz}$$

$$\le \tfrac{1}{2} N \max_i m_i G_i \cdot \|x_{n+1/2} - x_n\|^2 + \tfrac{1}{2} \max_i m_i G_i \cdot \|x_{n+1} - x_{n+1/2}\|^2 \qquad \text{(D.7)}$$

**Bounding (D.6b).** Again, by the Fenchel-Young inequality, we obtain:

$$\text{(D.6b)} \le \sum_{i \in \mathcal{N}} \frac{m_i(1 - \lambda_i)}{2G_i} \|v_i(x_n) - v_i(x_{n-1/2})\|_*^2 + \sum_{i \in \mathcal{N}} \frac{m_i(1 - \lambda_i)G_i}{2} \|x_{i,n+1} - x_{i,n+1/2}\|^2$$

$$\le \sum_{i \in \mathcal{N}} \frac{m_i(1 - \lambda_i)G_i}{2} \|x_n - x_{n-1/2}\|^2 + \sum_{i \in \mathcal{N}} \frac{m_i(1 - \lambda_i)G_i}{2} \|x_{i,n+1} - x_{i,n+1/2}\|^2$$

$$\text{\% } v_i(x) \text{ is } G_i\text{-Lipschitz}$$

$$\le \tfrac{1}{2} N \max_i m_i G_i \cdot \|x_n - x_{n-1/2}\|^2 + \tfrac{1}{2} \max_i m_i G_i \cdot \|x_{n+1} - x_{n+1/2}\|^2 \qquad \text{(D.8)}$$

**Bounding (D.6c).** Finally, by the lower bound on the Fenchel coupling of Proposition B.2, we get:

$$- \sum_{i \in \mathcal{N}} \frac{m_i}{\eta_i} F_i(x_{i,n+1}, y_{i,n+1/2}) - \sum_{i \in \mathcal{N}} \frac{m_i}{\eta_i} F_i(x_{i,n+1/2}, y_{i,n})$$

$$\le - \sum_{i \in \mathcal{N}} \frac{m_i K_i}{2\eta_i} \|x_{i,n+1} - x_{i,n+1/2}\|^2 - \sum_{i \in \mathcal{N}} \frac{m_i K_i}{2\eta_i} \|x_{i,n+1/2} - x_{i,n}\|^2 \qquad \text{\% by (B.21b)}$$

$$\le - \min_i \frac{m_i K_i}{2\eta_i} \cdot [\|x_{n+1} - x_{n+1/2}\|^2 + \|x_{n+1/2} - x_n\|^2] \qquad \text{(D.9)}$$

Thus, by folding Eqs. (D.7)–(D.9) back into (D.6), we obtain the bound

$$\sum_{i \in \mathcal{N}} m_i \langle v_i(x_{n+1/2}), p_i - x_{i,n+1/2}\rangle \le E_n - E_{n+1}$$

$$+ \frac{1}{2}\left(N \max_i m_i G_i - \min_i \frac{m_i K_i}{\eta_i}\right) \|x_{n+1/2} - x_n\|^2$$

$$+ \frac{1}{2}\left(2 \max_i m_i G_i - \min_i \frac{m_i K_i}{\eta_i}\right) \|x_{n+1} - x_{n+1/2}\|^2$$

$$+ \frac{1}{2} N \max_i m_i G_i \cdot \|x_n - x_{n-1/2}\|^2 \, . \qquad \text{(D.10)}$$

Now, if we instantiate (D.10) to $p \leftarrow q$ where $q$ is the strategic center of $\Gamma$, its left-hand side (LHS) will vanish by (HG-center). Hence, summing over all $n = 1, 2, \ldots, T$, (D.10) ultimately yields

$$0 \le E_1 + \frac{1}{2}\left(N \max_i m_i G_i - \min_i \frac{m_i K_i}{\eta_i}\right) \sum_{n=1}^T \|x_{n+1/2} - x_n\|^2$$

$$+ \frac{1}{2}\left((N + 2) \max_i m_i G_i - \min_i \frac{m_i K_i}{\eta_i}\right) \sum_{n=2}^T \|x_n - x_{n-1/2}\|^2$$

$$+ \frac{1}{2}\left(2 \max_i m_i G_i - \min_i \frac{m_i K_i}{\eta_i}\right) \|x_{T+1} - x_{T+1/2}\|^2$$

$$+ \frac{1}{2} N \max_i m_i G_i \cdot \|x_1 - x_{1/2}\|^2 . \tag{D.11}$$

Now, by our step-size assumption, we readily obtain

$$(N + 2) \max_i m_i G_i \leq \frac{1}{2} \min_i \frac{m_i K_i}{\eta_i} \tag{D.12}$$

so (D.11) becomes

$$0 \leq E_1 - \frac{1}{4} \min_i \frac{m_i K_i}{\eta_i} \sum_{n=1}^{T} \|x_{n+1/2} - x_n\|^2 - \frac{1}{4} \min_i \frac{m_i K_i}{\eta_i} \sum_{n=2}^{T} \|x_n - x_{n-1/2}\|^2 \tag{D.13}$$

where we used our initialization convention $x_1 = x_{1/2}$ and the fact that the third line of (D.11) is negative. We thus get

$$\sum_{n=1}^{T} \|x_{n+1/2} - x_n\|^2 + \sum_{n=2}^{T} \|x_n - x_{n-1/2}\|^2 \leq \frac{4E_1}{\min_i m_i K_i / \eta_i} \tag{D.14}$$

from which our assertion follows immediately. ∎

**D.2. Proof of Theorem 3.** We are now in a position to prove the regret guarantees of (FTRL+), which we restate below for convenience.

**Theorem 3.** *Suppose that each player in a harmonic game $\Gamma$ is following* (FTRL+) *with learning rate $\eta_i \leq m_i K_i [2(N + 2) \max_j m_j G_j]^{-1}$ and payoff models as per* (13a) *and* (15)*. Then the individual regret of each player $i \in \mathcal{N}$ is bounded as*

$$\mathrm{Reg}_i(T) := \max_{p_i \in \mathcal{X}_i} \sum_{n=1}^{T} [u_i(p_i; x_{-i,n}) - u_i(x_n)] \leq \frac{H_i}{\eta_i} + \frac{2G_i}{N + 2} \sum_{j \in \mathcal{N}} \frac{H_j}{\eta_j G_j} \tag{16}$$

*where $H_i = \max h_i - \min h_i$, and $G_i$ is the Lipschitz modulus of $v_i$.*

*Proof.* By a minor reshuffling of terms in (D.4), we readily get

$$\langle v_i(x_{n+1/2}), p_i - x_{i,n+1/2} \rangle \leq \frac{1}{\eta_i} [F_i(p_i, y_{i,n}) - F_i(p_i, y_{i,n+1})]$$

$$+ \langle v_i(x_{n+1/2}) - v_i(x_n), x_{i,n+1} - x_{i,n+1/2} \rangle$$

$$+ (1 - \lambda_i) \langle v_i(x_n) - v_i(x_{n-1/2}), x_{i,n+1} - x_{i,n+1/2} \rangle$$

$$- \frac{1}{\eta_i} F_i(x_{i,n+1}, y_{i,n+1/2}) - \frac{1}{\eta_i} F_i(x_{i,n+1/2}, y_{i,n}) \tag{D.15}$$

and hence, by a repeated application of the Fenchel-Young inequality in its Peter-Paul form:

$$\langle v_i(x_{n+1/2}), p_i - x_{i,n+1/2} \rangle \leq \frac{1}{\eta_i} [F_i(p_i, y_{i,n}) - F_i(p_i, y_{i,n+1})]$$

$$+ \frac{1}{2G_i} \|v_i(x_{n+1/2}) - v_i(x_n)\|_*^2 + \frac{G_i}{2} \|x_{i,n+1} - x_{i,n+1/2}\|^2$$

$$+ \frac{1 - \lambda_i}{2G_i} \|v_i(x_n) - v_i(x_{n-1/2})\|_*^2 + \frac{(1 - \lambda_i)G_i}{2} \|x_{i,n+1} - x_{i,n+1/2}\|^2$$

$$- \frac{K_i}{2\eta_i} \left[ \|x_{i,n+1} - x_{i,n+1/2}\|^2 + \|x_{i,n+1/2} - x_{i,n}\|^2 \right] . \tag{D.16}$$

Hence, by using the Lipschitz continuity of $v_i$, we finally get

$$\langle v_i(x_{n+1/2}), p_i - x_{i,n+1/2} \rangle \leq \frac{1}{\eta_i} [F_i(p_i, y_{i,n}) - F_i(p_i, y_{i,n+1})]$$

$$+ \frac{G_i}{2} \|x_{n+1/2} - x_n\|^2 + \frac{G_i}{2} \|x_{i,n+1} - x_{i,n+1/2}\|^2$$

$$+ \frac{G_i}{2} \|x_n - x_{n-1/2}\|^2 + \frac{G_i}{2} \|x_{i,n+1} - x_{i,n+1/2}\|^2$$

$$-\frac{K_i}{2\eta_i}\left[\|x_{i,n+1} - x_{i,n+1/2}\|^2 + \|x_{i,n+1/2} - x_{i,n}\|^2\right] \tag{D.17}$$

Thus, summing over $n = 1, 2, \ldots, T$, and keeping in mind that our assumptions for $\eta_i$ also give $G_i < K_i/(2\eta_i)$, we finally get

$$\sum_{n=1}^{T}\langle v_i(x_{n+1/2}), p_i - x_{i,n+1/2}\rangle \le \frac{H_i}{\eta_i} + \frac{G_i}{2}\left[\sum_{n=1}^{T}\|x_{n+1/2} - x_n\|^2 + \sum_{n=2}^{T}\|x_n - x_{n-1/2}\|^2\right] \tag{D.18}$$

where we used the fact that $F_i(p_i, 0) = h(p) - \min h_i \le \max h_i - \min h_i =: H_i$. Our result then follows by invoking (D.5) and using the fact that $m_i G_i \le \max_j m_j G_j$ for all $i \in \mathcal{N}$. ∎

**D.3. Proof of Theorem 4.** With all this in hand, we are finally in a position to prove our main equilibrium convergence result for (FTRL+). For convenience, we restate the relevant theorem below.

**Theorem 4.** *Suppose that each player in a harmonic game $\Gamma$ follows (FTRL+) with learning rate $\eta_i \le m_i K_i[2(N+2)\max_j m_j G_j]^{-1}$ and payoff models as per (13a) and (15). Then $x_n$ converges to the set of Nash equilibria of $\Gamma$.*

*Proof.* Our proof proceeds in a series of steps, as detailed below.

*Step 1: Convergence of energy levels.* We begin by showing that the energy $E_n \equiv E(q, y_n)$ of (FTRL+) relative to the game's harmonic center converges to some finite value $E_\infty$. That this is so follows from a well-known property of quasi-Fejér sequences [10, Lemma 3.1], whose proof we reproduce below for completeness.

Indeed, by Eq. (D.10) and Proposition D.2, we have

$$E_{n+1} \le E_n + \varepsilon_n \tag{D.19}$$

with $\varepsilon_n$, $n = 1, 2, \ldots$ summable. Letting $E'_n = E_n + \sum_{k=n}^{\infty} \varepsilon_k$, we further get

$$E'_{n+1} = E_{n+1} + \sum_{k=n+1}^{\infty} \varepsilon_k \le E_n + \sum_{k=n}^{\infty} \varepsilon_k = E'_n \tag{D.20}$$

by (D.19), so $E'_n$ converges. Since $\varepsilon_n$ is summable, it follows that $E_n$ also converges, as claimed. ◇

*Step 2: Boundedness of score differences.* We now proceed to show that the normalized score differences $z_n = \Pi(y_n)$ where $\Pi$ is the normalization operator (C.8) are bounded. Indeed, by the definition of $E_n = E(q, y_n) = \sum_{i \in \mathcal{N}}(m_i/\eta_i)F_i(q_i, y_{i,n})$, it follows that $\sup_n F_i(q_i, y_{i,n}) < \infty$ for all $i \in \mathcal{N}$. Thus, by Lemma C.5, we conclude that each component of $z_n$ is bounded, so $z$ is itself bounded. ◇

*Step 3: Convergent subsequences of (FTRL+).* We now observe that (FTRL+) enjoys the following series of properties:

1. The sequence $z_n = \Pi(y_n)$ admits a subsequence $z_{n_k}$ that converges to some limit point $z_\infty \in \mathcal{Z}$ (a consequence of the fact that $z_n$ is bounded, see above).

2. In turn, this implies that the subsequence $x_{n_k} = Q(y_{n_k}) = \hat{Q}(z_{n_k})$ converges to some $x_\infty \in \mathcal{X}$.

3. Since the sequences $A_n = \|x_{n+1/2} - x_n\|^2$ and $B_n = \|x_{n+1} - x_{n+1/2}\|^2$ are both summable (by Proposition D.2), we further have $\lim_{k\to\infty} x_{n_k+1/2} = x_\infty$ and, more generally, by a straightforward induction:

$$\lim_{k\to\infty} x_{n_k+r} = x_\infty \quad \text{for any (fixed) } r = 1/2, 1, 3/2, \ldots \tag{D.21}$$

4. Likewise, for the sequence of payoff signals $\hat{v}_n$, we have

$$\hat{v}_{i,n_k} = \lambda_i v_i(x_{n_k}) + (1 - \lambda_i) v_i(x_{n_k-1/2}) \xrightarrow[k\to\infty]{} \lambda_i v_i(x_\infty) + (1 - \lambda_i)v_i(x_\infty) = v_i(x_\infty) \tag{D.22}$$

so $\lim_{k\to\infty} v(x_{n_k}) = v(x_\infty)$. ◇

*Step 4: Variational characterization of limit points.* We now proceed to show that $v_\infty \coloneqq v(x_\infty)$ belongs to the polar cone $\mathrm{PC}(x_\infty) = \{w \in \mathcal{Y} : \langle w, x - x_\infty \rangle \leq 0 \text{ for all } x \in \mathcal{X}\}$ to $\mathcal{X}$ at $x_\infty$. To do so, suppose that (FTRL+) performs $r$ steps from $n_k$ so

$$y_{n_k+r} = y_{n_k} + \eta \sum_{j=1}^{r} \hat{v}_{n_k+1/2} \tag{D.23}$$

where, to ease notation, we have made the simplifying assumption that $\eta_i = \eta$ for all $i \in \mathcal{N}$.[13] Then, by invoking Lemma B.4 with $y \leftarrow y_{n_k}$ and $y^+ \leftarrow y_{n_k+r}$, we obtain

$$\left\langle \eta \sum_{j=1}^{r} \hat{v}_{n_k+1/2}, p - x_{n_k+r} \right\rangle \leq \langle \nabla h(x_{n_k+r}) - y_{n_k}, p - x_{n_k+r} \rangle$$

$$= \langle \nabla h(x_{n_k+r}) - z_{n_k}, p - x_{n_k+r} \rangle \tag{D.24}$$

where, in the second line, we have used the fact that $\langle y, x' - x \rangle = \langle \Pi(y), x' - x \rangle$ for all $x, x' \in \mathcal{X}$ and all $y \in \mathcal{Y}$. Thus, letting $k \to \infty$, we get from Step 3 and the continuity of $\nabla h$ that

$$\eta r \langle v(x_\infty), x - x_\infty \rangle \leq \langle \nabla h(x_\infty) - z_\infty, x - x_\infty \rangle \tag{D.25}$$

for all $r = 1, 2, \ldots$ and all $x \in \mathcal{X}$.[14] Since $r$ can be chosen arbitrarily, we must have $\langle v(x_\infty), x - x_\infty \rangle \leq 0$ for all $x \in \mathcal{X}$. Hence, by the variational characterization (VI) of Nash equilibria, we conclude that $x_\infty$ must be itself a Nash equilibrium of $\Gamma$, and our proof is complete. ∎

---

[13]This assumption does not affect the core of our arguments, but it greatly streamlines the presentation.

[14]The fact that $x_\infty \in \mathrm{dom}\, \partial h$ is a consequence of Lemma C.5 and the convergence of $E_n$ to $E_\infty < \infty$.

