# OpenReview forum: "No-regret Learning in Harmonic Games: Extrapolation in the Face of Conflicting Interests"
_NeurIPS.cc/2024/Conference — NeurIPS 2024 spotlight_

### Official Review · Reviewer_aEEw · 2024-07-09

**Soundness:** 4
**Presentation:** 4
**Contribution:** 4
**Rating:** 7
**Confidence:** 3

**Summary:**

The paper looks at how no-regret learning algorithms behave in harmonic games, which model situations where players have conflicting interests. This is different from the often-studied potential games where interests are shared. The authors show that in continuous time, FTRL dynamics are Poincaré recurrent and do not converge. In discrete time, FTRL can cause never-ending cycles of best responses. But, by adding an extrapolation step (FTRL+), they show that learning can reach a Nash equilibrium from any starting point, ensuring constant regret and giving important insights into harmonic games.

**Strengths:**

The paper stands out for its novel exploration of no-regret learning in harmonic games, which are less studied compared to potential games. The use of FTRL+ to ensure convergence in these games is a creative extension of existing algorithms. Furthermore, the the result on Poincare recurrence in Theorem 2 is insightful and interesting. The statements and proofs of the results are clear and rigorous, and the figured and presentation aid clarity.

**Weaknesses:**

Empirical results beyond the matching pennies example could strengthen the work. The proofs are somewhat dense and challenging to follow. A stronger differentiation from the work by Legacci et al. would be beneficial.

**Questions:**

How do the choices of parameters $\lambda_i$ and $\eta_i$ affect the performance of the FTRL+ algorithm? How might one select these parameters for different types of harmonic games? Could you expand on the types of harmonic games tested in your experiments? Are there benchmarks the algorithm could be compared against?

**Limitations:**

Limitations are sufficiently covered given the theoretical nature of the paper.

---

> ### Author Rebuttal · Authors · 2024-08-07
>
> Dear Reviewer aEEW,
>
> Thank you for your strong positive evaluation and encouraging remarks! We reply to your questions and comments below:
>
> > Empirical results beyond the matching pennies example could strengthen the work.
>
> Duly noted. We have included a pdf in our global rebuttal with additional figures in a series of $2\times 2 \times 2$ harmonic games that better illustrate the differences between FTRL and FTRL+.
>
>
> > The proofs are somewhat dense and challenging to follow.
>
> Point well taken. Being fully aware of the NeurIPS reviewing load and the time that it takes to review a paper, we tried to make our proofs as concise and straight-to-the-point as possible, but we understand that we may have overshot our original target. We will be happy to expand our proof sketch in the main part of the paper and provide more explanations and intuition along the way in the first revision opportunity.
>
>
> > A stronger differentiation from the work by Legacci et al. would be beneficial.
>
> To provide the necessary context, Legacci et al. [32] showed that the replicator dynamics – a special case of the FTRL dynamics with entropic regularization – are Poincaré recurrent in uniform harmonic games (a subclass of harmonic games for which the uniform distribution is always a Nash equilibrium). In the continuous-time setting, our paper's contribution is that *all* FTRL dynamics are recurrent in *all* harmonic games.
>
> The way that [32] obtained their result hinges on a special mathematical construct – the so-called *Shahshahani metric* – which is essentially "mandated" by the structure of the replicator dynamics. The key property of this metric is that incompressibility of the replicator field is equivalent to the underlying game being uniformly harmonic. As such, a plausible first step to extend the results and analysis of [32] would be to see if a suitably "adjusted" variant of the Shahshahani metric could provide an equivalence between incompressibility and harmonicity under a different measure; however, all our efforts to establish such an equivalence failed. Likewise, if we work with only *uniform* harmonic games, it is still not easy to find an analogue of the Shahshahani metric that provides a link between incompressibility of the associated FTRL dynamics and (uniform) harmonicity (at least, all our efforts to do so failed). Because of this, the "incompressibility" approach of [32] does not seem generalizable -- at least, not in a straightforward way.
>
> **TLDR:** We see no way of extending the incompressibility approach of [32] to more general FTRL dynamics and/or non-uniform harmonic games. We will be happy to include a version of this discussion when it is again possible to revise the paper to better highlight the differences between our dualization approach and the Riemannian incompressibility approach of [32].
>
>
> > How do the choices of parameters $\lambda_i$ and $\eta_i$ affect the performance of the FTRL+ algorithm? How might one select these parameters for different types of harmonic games?
>
> Great question! The main things to keep mind are as follows:
> - For $\eta_i$: in principle, $\eta_i$ should be chosen as large as possible, as larger values of $\eta$ lead to lower regret in (16), and to faster convergence to Nash equilibrium. The bound of L307 provides an upper limit to $\eta_i$ so, from an operational standpoint, $\eta$ should be taken as high as allowed by this limit.
> - For $\lambda_i$: the precise dependence is determined by Eq. (D.9) in Appendix D. We provided a rough upper bound at that point to simplify the end expression. In principle, taking $\lambda = 1$ leads to better bounds, but this comes at a cost of two queries per iteration, so there is still a trade-off involved (but one which is harder to quantify exactly).
>
> > Could you expand on the types of harmonic games tested in your experiments?
>
> The payoffs of each game appear in the vertices of the phase portrait (in the paper as well as the rebuttal pdf). The games themselves were generated by randomly selecting a game with integer payoffs from the subspace of games satisfying the harmonicity condition of Definition 1.
>
> > Are there benchmarks the algorithm could be compared against?
>
> The most widely used benchmarks that we are aware of are the three algorithms presented in Eqs. (14a,b,c) in our paper. Of these, (14a) is FTRL, which diverges (we plotted this throughout for comparison); (14b,c) are both special cases of FTRL+, and their trajectories are almost indistinguishable, so we only used the optimistic variant of the algorithm to minimize visual clutter.
>
> ---
>
> Please let us know if you have any follow-up questions, and thank you again for your time and positive evaluation!
>
> Kind regards,
>
> The authors

---

> > ### Comment · Reviewer_aEEw · 2024-08-13
> >
> > I thank the authors for their extensive response and clarifications. While I will leave my positive score of 7 as is, given the author's response I can increase my confidence.

---

> > > ### Author Response · Authors · 2024-08-14
> > > **Thank you**
> > >
> > > Thank you again for your time, input, and positive evaluation!
> > >
> > > Kind regards,
> > >
> > > The authors

---

### Official Review · Reviewer_sMJW · 2024-07-12

**Soundness:** 4
**Presentation:** 4
**Contribution:** 3
**Rating:** 6
**Confidence:** 4

**Summary:**

The paper studies the behavior of no-regret dynamics, and in particular follow the regularized leader (FTRL), in harmonic games--the strategic counterpart of potential games. They establish the following main results: i) the continuous-time version of FTRL is Poincare recurrent; ii) an extrapolated version of FTRL (FTRL+), which includes the optimistic and mirror-prox variant, converges to a Nash equilibrium; iii) under FTRL+ each player attains $O(1)$ regret.

**Strengths:**

The paper provides a clear and compelling motivation for investigating no-regret dynamics in harmonic games. In particular, the decomposition of Candogan et al. shows that, together with potential games, harmonic games constitute the basic building blocks of any game. Further, harmonic games generalized zero-sum games with a fully-mixed equilibrium, the latter class having received extensive attention in the literature. The results obtained in the paper make a concrete step towards better understanding no-regret dynamics in harmonic games, which I believe is an important contribution and will be welcomed by the learning in games community at NeurIPS. I also believe that the paper may lead to interesting follow-up work on harmonic games. Overall, the main message of the paper is compelling.

The writing of the paper is also exemplary and of really high quality. The main body does a great job at providing high-level sketches of the main technical ideas, and the appendix is also very well organized and polished. I did not detect any notable issues with the soundness of the claims.

**Weaknesses:**

Regarding the significance of the results, it can be argued that some of the results are somewhat incremental based on existing results. The fact that FTRL is Poincare recurrent is perhaps not that surprising conceptually given the recent paper of Legacci et al. which shows that for the special case of replicator dynamics; I do understand though that the extension requires some new ideas, as the authors discuss. Moreover, the fact that FTRL+ attains constant regret is also not particularly surprising. To put that result into better context, I would recommend citing the paper of Daskalakis, Fishelson and Golowich (NeurIPS 2021) that shows $O(\log^4 T)$ regret for any game; as far as I understand, the only improvement pertains to logarithmic factors in $T$, which is perhaps not very significant. But I do not believe that the points above constitute basis for rejection.

**Questions:**

No further questions.

**Limitations:**

The authors have adequately addressed the limitations.

---

> ### Author Rebuttal · Authors · 2024-08-07
>
> Dear Reviewer sMJW,
>
> Thank you for your positive evaluation and your input! We reply to your questions and remarks below:
>
> > The fact that FTRL is Poincare recurrent is perhaps not that surprising conceptually given the recent paper of Legacci et al. which shows that for the special case of replicator dynamics; I do understand though that the extension requires some new ideas, as the authors discuss.
>
> From a conceptual standpoint, we agree that it is plausible to expect that FTRL is recurrent in harmonic games, given that the replicator dynamics (a special case of FTRL dynamics) are recurrent in uniform harmonic games – the result of Legacci et al. [32]. At the same time, we were genuinely surprised that the proof technique of [32] could **not** be extended along either direction – that is, to prove that FTRL is recurrent in *uniform* harmonic games, or that the replicator dynamics are recurrent in *general* harmonic games.
>
> The reason for this is that the result of [32] revolves around a special mathematical construct – the so-called *Shahshahani metric* – which is essentially "mandated" by the choice of [32] to work with the replicator dynamics. The key property of this metric is that incompressibility / preservation of volume under the replicator dynamics is equivalent to the underlying game being uniformly harmonic. As such, a first step to extend the results of [32] would be to see if a suitably "adjusted" variant of the Shahshahani metric could provide an equivalence between incompressibility and harmonicity under a different measure – however, all our efforts to establish such an equivalence failed. Likewise, if we work with only *uniform* harmonic games, it is still not easy to find an analogue of the Shahshahani metric that makes a given, non-replicator FTRL scheme incompressible (at least, all our efforts to do so failed). Because of this, the "incompressibility" approach of [32] does not seem generalizable – at least, not in a straightforward way.
>
> **TLDR:** Even though, conceptually speaking, the recurrence of FTRL may not seem surprising, the failure of the incompressibility approach of [32] in more general settings and the apparent need to take a radically different approach does seem surprising (at least to us), and we believe that it carries important insights as to which techniques are more suitable for the analysis of general harmonic games.
>
>
> > Moreover, the fact that FTRL+ attains constant regret is also not particularly surprising. To put that result into better context, I would recommend citing the paper of Daskalakis et al, that shows $\log^4 T$ regret for any game.
>
> First off, thanks for bringing up the paper of Daskalakis et al! We should have cited and discussed this paper, this was an omission on our end.
>
> Now, as to the significance of the polylog factor: if $T$ ranges between $10^3$ and $10^6$ (a reasonable range for online learning applications), the factor $\log^4 T$ ranges roughly between $2000$ and $35000$. Thus, ceteris paribus, shaving off the $\log^4 T$ factor could result in a gain between $3$ and $4$ orders of magnitude, depending on the context. We believe that this gain can be significant in several applications – though we freely acknowledge that there is a certain degree of subjectivity as to which gains are considered significant in which setting.
>
> Going beyond this point, we should clarify that, in our view, the true importance of the constant regret guarantee of FTRL+ is the associated path length bound (D.6) which, in turn, plays a crucial part in establishing the convergence of FTRL+ to equilibrium. This would not be possible with a non-constant bound (at least, not without a fairly different technical approach), a fact which, from our standpoint, further adds to the significance of an $\mathcal{O}(1)$ versus an $\mathcal{O}(\mathrm{polylog} T)$ bound.
>
> ---
>
> Please let us know if you have any follow-up questions, and thank you again for your time and positive evaluation!
>
> Kind regards,
>
> The authors

---

> > ### Comment · Reviewer_sMJW · 2024-08-11
> >
> > I thank the authors for the detailed response. I have no further questions, and I maintain my positive evaluation.

---

> > > ### Author Response · Authors · 2024-08-12
> > > **Thank you**
> > >
> > > Thank you again for your time, input, and positive evaluation!
> > >
> > > Kind regards,
> > >
> > > The authors

---

### Official Review · Reviewer_gj66 · 2024-07-13

**Soundness:** 4
**Presentation:** 3
**Contribution:** 3
**Rating:** 7
**Confidence:** 4

**Summary:**

This paper studies multi-agent no-regret learning dynamics in general harmonic games, which is the strategic complement of potential games. The paper's main contributions are the convergence properties of the family of "Follow-the-regularized-leader" (FTRL) algorithms and its variants in general harmonic games, significantly extending previous results in two-player zero-sum games with fully mixed equilibrium and uniformly harmonic games. Specifically, the main results are (1) the continuous-time dynamics of FTRL are Poincaré recurrent, (2) the discrete-time dynamics of FTRL diverges, but FTRL with an extrapolation step (called FTRL+) guarantees $O(1)$ individual regret and global asymptotic last-iterate convergence/point convergence to a Nash equilibrium. These results show that potential and harmonic games are complementary not only from the strategic but also from the dynamic viewpoint.

**Strengths:**

1. This paper is very well-written and easy to follow. I appreciate that the main body clearly explains the main ideas, and the appendix provides rigorous and detailed proof.
2. The problem of no-regret learning dynamics in games is relevant. This paper contributes to the area by establishing the previously unknown convergence properties of the FTRL dynamics in general harmonic games, including Poincaré recurrence of continuous-time FTRL, last-iterate convergence, and constant regret of FTRL+. The results and techniques in this paper shed light on the further development of learning in harmonic games.

**Weaknesses:**

I do not see any major weakness in the paper.

Minor comments:

Theorem 3/4: $m_i$ is used to choose the step size but has not been defined? The definition is in the appendix but a pointer should be given in the main body.

Line 314: "Similar bounds have only been established for optimistic methods in two-player zero-sum games [25]". [25] established constant regret bounds for all variationally stable games, a class of multi-player games that includes two-player zero-sum games. This should be acknowledged.

[25] Hsieh, Yu-Guan, Kimon Antonakopoulos, and Panayotis Mertikopoulos. "Adaptive learning in continuous games: Optimal regret bounds and convergence to nash equilibrium." In Conference on Learning Theory. 2021

**Questions:**

1. $m_i$ is used to choose the step size for convergence of FTRL+. Is there an efficient way to estimate an upper bound of $m_i$?
2. This paper left the rate of convergence for FTRL+ as an open question. I would like to know if similar results hold for optimistic online mirror descent (OOMD)-type algorithms in harmonic games. I think proving convergence rates for OOMD algorithms, especially OGDA, might be more promising than FTRL+.

---

> ### Author Rebuttal · Authors · 2024-08-07
>
> Dear Reviewer gj66,
>
> Thank you for your strong positive evaluation and encouraging remarks! We reply to your questions and comments below:
>
> > Theorem 3/4: $m_i$ is used to choose the step size but has not been defined? The definition is in the appendix but a pointer should be given in the main body.
>
> $m_i$ is actually defined in L140, just after the definition of a harmonic game, but we understand that it is easy to miss. We will nest it inside Definition 1 to make it easier to find – thanks for pointing this out!
>
>
> > "Similar bounds have only been established for optimistic methods in two-player zero-sum games [25]". [25] established constant regret bounds for all variationally stable games, a class of multi-player games that includes two-player zero-sum games. This should be acknowledged.
>
> Fair point. This was an omission on our end, we will rectify it accordingly, thanks for pointing it out!
>
>
> > $m_i$ is used to choose the step size for convergence of FTRL+. Is there an efficient way to estimate an upper bound of $m_i$?
>
> Good point! In general, we do not see an easy way of obtaining an upper bound for $m_i$ based on *local* knowledge alone. However, as far as the step-size tuning is concerned, we believe that an AdaGrad-type step-size would obviate the need to estimate $m_i$, as it obviates the need to know the Lipschitz modulus of the players' payoff functions in zero-sum games. As we state in the paper's concluding remarks, this is a very fruitful direction for future research, and one which we intend to undertake in future work.
>
>
> > This paper left the rate of convergence for FTRL+ as an open question. I would like to know if similar results hold for optimistic online mirror descent (OOMD)-type algorithms in harmonic games. I think proving convergence rates for OOMD algorithms, especially OGDA, might be more promising than FTRL+.
>
> We are not aware of *any* convergence rates for harmonic games (or even asymptotic convergence results for that matter); to the best of our knowledge, our paper provides the first convergence result in the literature for general harmonic games.
>
> Now, we agree that optimistic GD might be easier to analyze than FTRL+ in terms of rates, and it would be a natural first step. However, when the players' regularizer function is steep (in the sense that the induced choice map takes values in the interior of the strategy simplex), optimistic FTRL and optimistic mirror descent are equivalent, so all the difficulty would already be present in the OptMD case. Still, beyond the "steep" regime, it may well be that the "primal-dual" formulation of mirror descent (versus that of dual averaging) could be more amenable to a rate analysis. At the current state of our knowledge for learning in harmonic games, it is very difficult to tell without doing the full analysis.
>
> ---
>
> Thank you again for your time and positive evaluation – and please let us know if you have any follow-up questions!
>
> Kind regards,
>
> The authors

---

> > ### Comment · Reviewer_gj66 · 2024-08-09
> > **Response by the Reviewer**
> >
> > Thank you for the response! I agree that developing algorithms that require no knowledge of $m_i$ is an important question.  I will keep my score.
> >
> > Additional Comment: I think providing more examples of Harmonic games would be helpful. Are there other natural families of games (beyond that obtained by the decomposition or 2p0s game with fully mixed NE) that are harmonic?

---

> > > ### Author Response · Authors · 2024-08-12
> > >
> > > Thank you for your continued input and support!
> > >
> > > Regarding your question: other natural familes of games that are harmonic include the class of cyclic games (Hofbauer & Schlag, 2000), the Dawkins variants of the Battle of the Sexes (Smith & Hofbauer, 1987), crime-deterrence games (Cressman & Morrison, 2000), etc. [To be clear, these families of games predate the introduction of the term "harmonic game" in the literature (which was due to Candogan et al., 2011), but they were all seen to be harmonic once the notion was introduced]
> > >
> > > We will be sure to include these examples in the first revision opportunity, thanks for bringing up the question!
> > >
> > > Kind regards,
> > >
> > > The authors
> > >
> > > ---
> > >
> > > ### **References**
> > >
> > > - R. Cressman and W.G. Morrison. *On the evolutionary dynamics of crime.* The Canadian Journal of Economics/Revue canadienne d’Economique, 31(5):1101–1117, 1998.
> > > - J. Hofbauer and K.H. Schlag. *Sophisticated imitation in cyclic games.* Journal of Evolutionary Economics, 10(5):523–543, 2000.
> > > - J.M. Smith and J. Hofbauer. *The “battle of the sexes”: A genetic model with limit cycle behavior.* Theoretical population biology, 32(1):1–14, 1987.

---

### Official Review · Reviewer_1Wrk · 2024-07-15

**Soundness:** 3
**Presentation:** 3
**Contribution:** 3
**Rating:** 7
**Confidence:** 5

**Summary:**

The contributions of this paper are two-fold:

i) They prove that continuous FTRL dynamics for general harmonic games are Poincaré recurrent and hence do not converge. Their result generalizes the original result of

Mertikopoulos, P., Papadimitriou, C. H., and Piliouras, G., 'Cycles in adversarial regularized learning,' In SODA ’18: Proceedings of the 29th annual ACM-SIAM Symposium on Discrete Algorithms, 2018,

for zero-sum games with an interior equilibrium (as a trivial example of harmonic games), and the result of

Legacci, Davide, Panayotis Mertikopoulos, and Bary Pradelski, 'A geometric decomposition of finite games: Convergence vs. recurrence under no-regret learning,' arXiv preprint arXiv:2405.07224 (2024),

for uniform harmonic games to general harmonic games by volume-preserving flow arguments.

ii) Moreover, they show that the regret of general harmonic games for the class of Extrapolated FTRL algorithms (discrete-time dynamics) is constant in time ($T$).

**Strengths:**

- The contributions of this paper are solid and interesting. This paper addresses fundamental problems in harmonic games.

- I enjoyed reading the paper. It is well-written, especially the introduction to harmonic games in the appendix and the new viewpoint of Mixed characterization of harmonic games and its role on upper bounding the path length of the no-regret dynamics.

**Weaknesses:**

- Just some missing citations on no-regret learning for games, e.g.,

-- Daskalakis, Constantinos, Alan Deckelbaum, and Anthony Kim. "Near-optimal no-regret algorithms for zero-sum games." Proceedings of the twenty-second annual ACM-SIAM symposium on Discrete Algorithms. \
-- Chen, Xi, and Binghui Peng. "Hedging in games: Faster convergence of external and swap regrets." Advances in Neural Information Processing Systems 33 (2020). \
-- Piliouras, Georgios, Ryann Sim, and Stratis Skoulakis. "Beyond time-average convergence: Near-optimal uncoupled online learning via clairvoyant multiplicative weights update." Advances in Neural Information Processing Systems 35 (2022). \
...


__Minor Suggestions__
- Please emphasize in the claims that the regret bounds entailed are constant in $T$ and not potentially in the dimension of the action sets $\mathcal{A}$, since $H_i$ is not constant in $|\mathcal{A}_i|$.
- Regarding the notation for $\nu_i(x)$, maybe consider changing it to $\nu_i(x_{-i})$ to improve readability.
- In line 268, $x_{n + 1}$ instead of $x_{n}$?
- In line 277, "and" before "which" seems to be a typo.

**Questions:**

I do not have any particular question (as the results shown in this paper are not unexpected) except that could authors provide some intuition on the conjecture that convergence to Nash Eq. of no-regret learnings for general harmonic games would be linear beyond the work of

Wei, C.-Y., et al. Linear last-iterate convergence in constrained saddle-point optimization. In ICLR ’21: Proceedings of the 2021 International Conference on Learning Representations, 2021.

? (Non-asymptotic version of Theorem 4)

**Limitations:**

Not applied.

---

> ### Author Rebuttal · Authors · 2024-08-07
>
> Dear Reviewer 1Wrk,
>
> Thank you for your strong positive evaluation and encouraging remarks! We reply to your questions and comments below:
>
> > Just some missing citations on no-regret learning for games [suggestions follow]
>
> Thanks for the pointers, we were aware of some but not all of the references you provided -- we will discuss them accordingly.
>
>
> > Please emphasize in the claims that the regret bounds entailed are constant in $T$ and not potentially in the dimension of the action sets $A$, since $H$ is not constant in $A$.
>
> Duly noted - we will note this dependence explicitly to avoid any misunderstandings. Thanks for bringing this up!
>
>
> > Regarding the notation for $v_i(x)$, maybe consider changing it to $v_i(x_{-i})$ to improve readability.
>
> You mean in order to make explicit the non-dependence on $x_i$, correct? This is an excellent suggestion - we will run through the paper with a fine-toothed comb to make sure it does not create any clashes of notation and adjust things accordingly. Thanks!
>
> > [Typos]
>
> Will fix those, many thanks for the careful read!
>
>
> > Could the authors provide some intuition on the conjecture that convergence to Nash Eq. of no-regret learnings for general harmonic games would be linear beyond the work of [Wei et al.]? (Non-asymptotic version of Theorem 4)
>
> Our intuition comes from the fact that the harmonic measure always gives rise to a fully mixed equilibrium. Since this *specific* equilibrium of the game lies in the interior of the simplex, the weighted sum in the definition of a harmonic game formally looks quite similar to the condition needed to establish metric subregularity in the work of [Wei et al.]. Our conjecture reflects our (optimistic :-) ) belief that this formal similarity could be leveraged to yield a linear convergence rate result in the spirit of [Wei et al.]. However, at this stage, there are too many unknown variables - and metric subregularity can be tricky in problems without a convex structure to rely on - so, for the moment, this is purely conjectural.
>
> ---
>
> Please let us know if you have any follow-up questions, and thank you again for your time and positive evaluation!
>
> Regards,
>
> The authors

---

> ### Comment · Reviewer_1Wrk · 2024-08-08
> **Discussion on Log T for general sum games**
>
> Thank you very much for your detailed reply, especially regarding the intuition behind possible linear convergence.
>
> __This paper makes a solid and clear contribution, which is not surprising as harmonic games are a potential generalization of zero-sum games. Therefore, it was expected to observe constant regret. As a result, I would like to keep my score (accept) and thank the authors for their good work.__
>
> Lastly, I would like to ask the authors to please add a discussion on how their work improves the log/poly log $T$ for general-sum games, referencing the works of:
>
> - Piliouras, Georgios, Ryann Sim, and Stratis Skoulakis. "Beyond time-average convergence: Near-optimal uncoupled online learning via clairvoyant multiplicative weights update." Advances in Neural Information Processing Systems 35 (2022).
> - Daskalakis, Constantinos, Maxwell Fishelson, and Noah Golowich. "Near-optimal no-regret learning in general games." Advances in Neural Information Processing Systems 34 (2021): 27604-27616.
> - Farina, Gabriele, et al. "Near-optimal no-regret learning dynamics for general convex games." Advances in Neural Information Processing Systems 35 (2022): 39076-39089.
>
> to constant regret in $T$ only for harmonic games as a special class of general-sum games. This will clarify the scope of the contribution of this work and align it more closely with the existing literature.

---

> > ### Author Response · Authors · 2024-08-12
> > **Thank you**
> >
> > Thank you for the added pointers on $\log T$ regret, they are very helpful! Thank you also for your time, your continued input and support, all greatly appreciated.
> >
> > Kind regards,
> >
> > The authors

---

### Author Rebuttal · Authors · 2024-08-07

Dear reviewers, dear AC,

We are grateful for your time, comments, and positive evaluation!

To streamline the discussion phase, we replied to each of your questions and comments in a separate rebuttal below, and we will integrate all applicable points in the next revision opportunity. To better illustrate the differences between FTRL and FTRL+, we have included in this global reply a pdf with further experiments for $2\times 2\times 2$ games, with different initializations in different harmonic games.

Thank you again for your input and encouraging remarks, and we are looking forward to continuing this constructive exchange during the discussion phase.

Kind regards,

The authors

---

### Decision · Program_Chairs · 2024-09-25

**Decision:**

Accept (spotlight)

**Comment:**

This paper considers the problem of no-regret learning in games, for a general class of *harmonic* games which can be viewed as generalizing zero-sum games. The main results are to (1) show that in continuous time, FTRL dynamics in harmonic games are Poincare recurrent and do not converge, and (2) a variant of FTRL with extrapolation can achieve O(1) regret for general harmonic games (improving recent log(T) regret results for general-sum games).

Reviewers unanimously agreed on acceptance, and felt that the paper makes a solid and clear contribution that meaningfully expands our understanding of no-regret learning dynamics, highlighting the role of harmonic games as a counterpart to potential games.

For the final version of the paper, please incorporate the reviewers' suggestion to include a comparison to recent works that achieve log(T) regret in general-sum games.